



# Comparison of Holocene temperature reconstructions based on GISP2 multiple-gas-isotope measurements

Michael Döring[1,2*] and Markus Christian Leuenberger[1,2]

[1]Climate and Environmental Physics, University of Bern, Switzerland
[2]Oeschger Centre for Climate Change Research (OCCR), Bern, Switzerland

*Correspondence to:* Michael Döring (doering@climate.unibe.ch)

Keywords: temperature reconstruction, ice core, nitrogen isotope, argon isotope, inverse-model, firn-model, accumulation-rate

**Abstract:** Nitrogen and argon stable-isotope data extracted from ancient air in ice cores provides the possibility
to reconstruct Greenland past temperatures when inverting firn-densification and heat-diffusion models (firn-models) to fit the gas-isotope data ($\delta^{15}N$, $\delta^{40}Ar$, $\delta^{15}N_{excess}$). This study uses the Döring and Leuenberger (2018) fitting-algorithm coupled on two state of the art firn-models to fit multiple Holocene gas-isotope data measured on the GISP2 ice core. We present for the first time the resulting temperature estimates when fitting $\delta^{15}N$, $\delta^{40}Ar$ and $\delta^{15}N_{excess}$ as single targets with misfits generally in the low permeg level. Whereas the comparison between
the reconstructions using $\delta^{15}N$ and $\delta^{40}Ar$ shows a high agreement, the use of $\delta^{15}N_{excess}$ for reconstructing temperature is problematic due to higher statistical and systematic data uncertainty influencing especially multi-decadal to multi-centennial signals and results in an unrealistic temperature estimate that differs significantly from the two other reconstructions. We find evidence for systematic too high $\delta^{40}Ar$ data in the early- and late-Holocene potentially caused by post coring gas-loss or an insufficient correction of this mechanism. Next, we
compare the performance of the Goujon et al. (2003) firn-model and the Schwander et al. (1997) firn-model for Holocene temperature reconstructions. Besides small differences of the reconstructed temperature anomalies – potentially caused by slightly different implementation of firn physics and parameters in the two models – the reconstructed temperature anomalies are highly comparable. We were able to quantify the contribution of the firn-model difference to the uncertainty budget of our reconstruction. Furthermore the fractions of uncertainty on
the reconstructed temperatures arising from the non-perfect reproducibility of the fitting algorithm and from the remaining final misfits (low permeg level) were quantified. Together with the published measurement uncertainty of the gas-isotope data and the analysis of the impact of accumulation-rate uncertainty on the reconstruction we were able to calculate the mean uncertainty ($2\sigma$) for the nitrogen and the argon based temperature estimates with $2\sigma_T = 0.80…0.88$ K for $T(\delta^{15}N)$, and $2\sigma_T = 0.87…1.81$ K for $T(\delta^{40}Ar)$, respectively.
Finally, we compare our reconstructed temperatures to two recent reconstructions based on the same gas-isotope data as used here, but following different reconstruction strategies: first the study of Buizert et al. (2018), which uses a combination of $\delta^{18}O_{ice}$-calibration and $\delta^{15}N$-fitting, and second the study of Kobashi et al. (2017), where $\delta^{15}N_{excess}$ was fitted in order to conduct the temperature reconstruction. We find generally higher agreement between our $T(\delta^{15}N)$ estimate and the Buizert et al. (2018) temperature – in terms of variability and correlation in
three investigated periodic-time bands (multi-decadal, multi-centennial and multi-millennial) – as if our $T(\delta^{15}N)$ reconstruction is compared to the Kobashi et al. (2017) temperature. However, all three reconstruction strategies lead to distinct temperature realizations.



## 1 Introduction

The use of nitrogen and argon stable-isotope variations in air extracted from ice cores is a relatively new tool for reconstructing past temperature (e.g. Huber et al., 2006b; Kindler et al., 2014; Kobashi et al., 2011; Landais et al., 2006; Orsi et al., 2014; Severinghaus et al., 1998, 2001). This method uses the stability of isotopic

compositions of nitrogen and argon in the atmosphere at orbital timescales as well as the fact that changes are only driven by processes in polar firn (Leuenberger et al., 1999; Mariotti, 1983; Severinghaus et al., 1998) and provides an alternative to the classical calibration of the stable oxygen ($\delta^{18}$O) and hydrogen isotopes extracted from the ice-core-water samples (Gierz et al., 2017; Johnsen et al., 2001; Steen-Larsen et al., 2011; Stuiver et al., 1995). The isotopic composition of the water samples provides a rather robust proxy for reconstructing paleo-

temperatures for times where large temperature variations occur (Gierz et al., 2017). In the Holocene where temperature variations are comparatively small, changes in seasonal distribution of precipitation as well as of evaporation conditions at the source region may dominate water-isotope-data variations (Huber et al., 2006b; Kindler et al., 2014; Werner et al., 2001). Recent studies (Buizert et al., 2018; Kobashi et al., 2017) used the nitrogen and argon isotopes of the GISP2 ice core (Greenland Ice Sheet Project Two, Meese et al., 1994;

Rasmussen et al., 2008; Seierstad et al., 2014) to reconstruct Holocene temperature variations for Greenland summit following different reconstruction strategies. Kobashi et al. uses the second-order parameter $\delta^{15}$N$_{excess}$ together with the firn-densification and heat-diffusion model from Goujon et al. (2003) to obtain a Holocene temperature estimate. Buizert et al. reconstructed summit temperatures by calibrating the GISP2 $\delta^{18}$O data by forcing the temperature to reproduce the general trend in $\delta^{15}$N using a dynamical firn-model. Both methods lead

to different temperature estimates. In the Buizert et al. study an overall uncertainty of 1.5 K was stated for the reconstructed temperature. Kobashi et al. estimated the uncertainty of the temperature reconstruction by examining the variance of temperature realizations when shifting the $\delta^{15}$N$_{excess}$ data in the range of analytical uncertainty before using his fitting approach. This approach results in an averaged uncertainty of 1.21 K (1σ). Döring and Leuenberger (2018) showed an automated approach enabling fitting gas-isotope data with an

outstanding accuracy with mismatches generally below the analytical uncertainty of the isotope measurements. It was shown on synthetic data experiments that in the case of perfectly known accumulation-rate data and neglecting noise, the remaining mismatches would lead to a temperature uncertainty (2σ) below 0.3 K for a single measurement and Holocene-like conditions. This study focuses on the challenges of temperature reconstructions using gas-isotope fitting for real Holocene data. We will discuss different aspects which are in

our view integral for the evaluation of the correctness of the reconstructed temperature estimates. First, we will discuss the gas-isotope data in the context of measurement uncertainty and focus on the suitability of the different isotopic quantities for reconstructing robust temperature estimates. Next, we discuss the reproducibility and the contribution of the final misfits to the uncertainty budget using our fitting approach. Also we will show the influence of different accumulation-rate estimates on the temperature reconstruction. Finally, we compare the

temperature solutions obtained by fitting $\delta^{15}$N, $\delta^{40}$Ar and $\delta^{15}$N$_{excess}$ to each other and place them in a context to the estimates of Kobashi et al. (2017) and Buizert et al. (2018). For our reconstruction we used two different firn-models, the models of Schwander et al. (1997) and of Goujon et al. (2003). We will compare our results using both of them and, we will provide an overall uncertainty of our method for the most robust estimate using all available information.





## 2 Data and method

### 2.1 Firn-models and inversion algorithm

The observed gas-isotope data mainly relies on firn densification processes combined with gas and heat diffusion (Severinghaus et al., 1998). So the use of firn-densification and heat-diffusion models (from now on referred to as firn-model) describing the physics of densification and heat and gas transport are necessary for inverting

measured gas-isotope data to surface temperatures. Also, accurate accumulation-rate data are needed to drive those models. In this study we use two commonly used firn-models. The first model was developed by Schwander et al. (1997) and used for the temperature reconstructions by Huber et al. (2006b) and Kindler et al. (2014). The second one, the model from Goujon et al. (2003) (adapted to the GISP2 site for this study), was used

i.a. in the studies by Guillevic et al. (2013), Kobashi et al. (2015), and Kobashi et al. (2017). While both models differs in the realisation of the well mixed zone at the top of the firn layer (convective zone) and the geothermal heat-flux throughout the ice sheet, both models are highly comparable and their differences correspond in first order to a constant offset in absolute temperature but leave temperature anomalies unchanged, as it was shown by other studies (e.g. Guillevic et al., 2013, Fig. 3) for glacial conditions. This is especially true for high

accumulation sites and time intervals with low amplitudes of temperature changes such as the Holocene investigated in this study. As a comparison of temperature reconstructions for Holocene conditions using both models is done for the first time in this study, we will again prove the comparability of the solutions gained by using both models. The conversion of gas-isotope data to surface temperature estimates using a firn-model is an inverse problem. The firn-model acts as a non-linear transfer function, combining temperatures and

accumulation-rates with the gas-isotope data. To solve this problem we use the automated fitting algorithm developed by Döring and Leuenberger (2018). The algorithm minimizes the mismatches between modelled (using a firn-model) and measured gas-isotope data using a combination of a Monte-Carlo based iterative approach and the analysis of remaining mismatches between modelled and target data (gas-isotope data). It is based on cubic-spline filtering (Enting, 1987) of random numbers and the laboratory-determined thermal-

diffusion-sensitivities for gas-isotopes. The used algorithm allows the fitting of the gas-isotope data with misfits in the low permeg level, mainly below the analytic measurement uncertainties. For modelling $\delta^{40}Ar$ and $\delta^{15}N_{excess}$ data we use, in addition to the details presented in Döring and Leuenberger (2018) the thermal-diffusion constant $\alpha_{T,Ar}$ and thermal-diffusion sensitivity $\Omega_{Ar}$ that have been empirically derived by Grachev and Severinghaus (2003):

$$\alpha_{T,Ar}(t) = \left(26.08 - \frac{3952}{\overline{T}(t)}\right) \cdot 10^{-3} \tag{1}$$

$$\Omega_{Ar}(t) = \frac{\alpha_{T,Ar} \cdot 10^3}{\overline{T}(t)} = \frac{26.08}{\overline{T}(t)} - \frac{3952}{\overline{T}(t)^2} \tag{2}$$

$\overline{T}(t)$ is the mean firn temperature (Leuenberger et al., 1999).

*The Schwander et al. (1997) firn-model*

The Schwander et al. (1997) firn-model is a semi-empirical model. It calculates the densification of the firn as well as heat- and gas-diffusion in the firn. The densification of the firn is calculated in three density intervals. In the density range of 345 kg m$^{-3}$ to 550 kg m$^{-3}$ the Herron-Langway model (Herron and Langway, 1980) is used.





The second interval (550 kg m$^{-3}$ to 800 kg m$^{-3}$) is calculated according to Barnola et al. (1991) and for $\rho > 800$ kg m$^{-3}$ is calculated after Wilkinson and Ashby (1975). The density threshold where the air bubbles are closed by the surrounding ice is implemented using the empirical formula from Martinerie et al. (1994) with a slight correction of -14 kg m$^{-3}$. Heat diffusion in the firn is obtained by using a simplified form of Paterson

(1994) where horizontal heat diffusion and internal heat production are neglected. Up to 300 m depth, heat diffusion is calculated with annual layer resolution and from 300 m to 1300 m the depth resolution is 100 m. Also at 1300 m the heat flux is defined to be zero. The molecular diffusion of air in the firn column is obtained by a one-dimensional box model (Schwander et al., 1993). It has to be mentioned again that no geothermal heat-flux emerging from the bottom of the ice sheet is implemented in Schwander et al. (1997) model. Also no

convective zone at the top of the firn layer is defined in the model.

*The Goujon et al. (2003) firn-model*

The Goujon et al. (2003) firn-model is a derivative of the densification model by Arnaud et al. (2000). A time-step of one year is used for calculating densification and heat diffusion. The spatial resolution is 25 cm in the top 150 m and decreases with depth where it reaches 25 m at bedrock. Heat diffusion in the ice is calculated across

the entire ice sheet (from bedrock to surface) using a simplified version of the heat-diffusion-model from Ritz (1989) where horizontal heat diffusion and internal heat production are neglected. The ice temperature at the bedrock is fixed with a constant value, which can be changed in the code to adapt the model for a specific site. For the GISP2 location a value of -4.5°C was used. In contrast to the Schwander et al. (1997) model, the Goujon et al. (2003) model provides the possibility to obtain a rough estimate of the present-day borehole temperature

profile, which can be used as an additional constrain for temperature reconstruction (see sect. 3.2.3.). The Goujon et al. (2003) firn-model also includes a constant (in time) convective zone which can be adapted in the code for a specific ice core location. A value of 2 m was used for the GISP2 site. The size of the convective zone affects the height of the diffusive firn column and therefore the enrichment of $\delta^{15}$N and $\delta^{40}$Ar caused by gravitational settling (Craig et al., 1988; Schwander, 1989). However, heat diffusion still occurs in the

convective zone.

**2.2 Timescale, and necessary data**

*Timescale*

For the entire study, the GICC05 chronology is used (Rasmussen et al., 2014; Seierstad et al., 2014). Following Döring and Leuenberger (2018), the temperature input is split into two parts in time. The first part ranges from

10.5 kyr to 35 kyr b2k ("spin-up section"), whereas the second part ranges from 0.02 kyr to 11.5 kyr b2k ("reconstruction window") for which we allow the fitting algorithm to change the temperature. The accumulation-rate as well as the surface temperature of the spin-up section remains unchanged during the reconstruction procedure.

*Target data*

The GISP2 gas-isotope data measured by Kobashi et al. (2008b) will be used for the Holocene temperature reconstruction as targets for our fitting algorithm (Döring and Leuenberger, 2018). Figure 1 shows the different gas-isotope targets ($\delta^{15}$N, $\delta^{40}$Ar, $\delta^{15}$N$_{excess}$) set on the GICC05 chronology. In tab.1 we list measurement uncertainties for the target data given in Kobashi et al. (2008b). As different uncertainties were given in the





Kobashi et al. publication for different measurement campaigns we consequently use the minimal and maximal uncertainties given in Kobashi et al. (2008b) for a signal-to-noise analysis of these data (see sect. 3.1.).

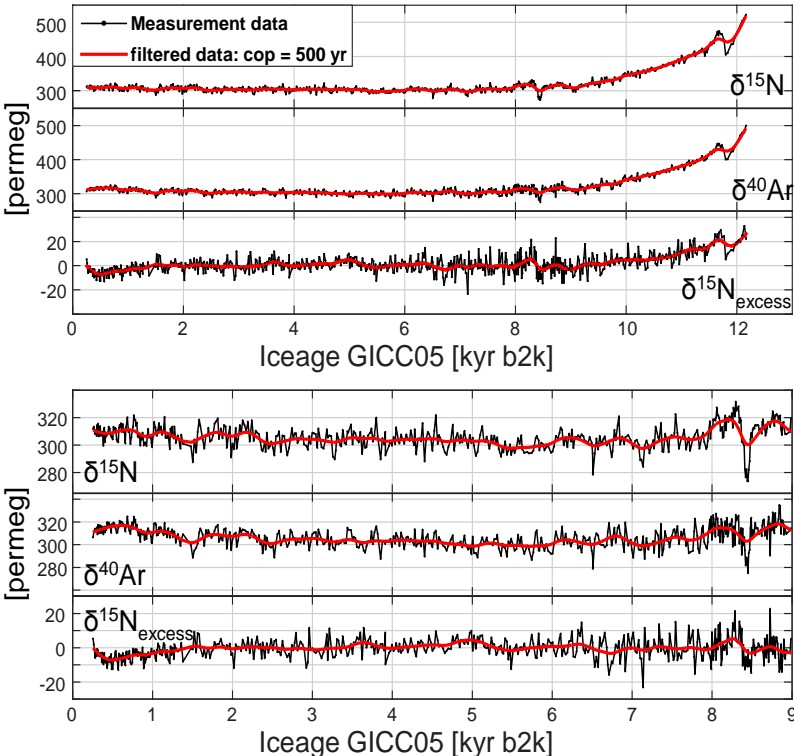

5  **Figure 1: Gas-isotope target data on GICC05 time scale (Kobashi et al., 2008b): Upper three plots: full δ15N, δ40Ar/4 and δ15Nexcess time-series; Bottom three plots: zoom in for the recent 9 kyr time window of the same quantities.**

| Target: | $\delta^{15}N$ | | $\delta^{40}Ar/4$ | | $\delta^{15}N_{excess}$ | |
|---|---|---|---|---|---|---|
| | max | min | max | min | max | min |
| Data uncertainty 1σ [permeg]*: | 4.0 | 3.0 | 9.0 | 4.0 | 9.8 | 5.0 |
| Signal uncertainty 1σ [permeg]: | 5.7 | 4.2 | 12.7 | 5.7 | 13.9 | 7.1 |
| SNR > 1 hf-signals [%]: | 70.0 | 78.0 | 35.7 | 74.2 | 16.5 | 52.3 |

**Table 1: Reported data uncertainty (*Kobashi et al., 2008b), calculated signal uncertainties (see text) and percentage of high frequency signals with SNR > 1.**





*Accumulation-rate input data*

As in Döring and Leuenberger (2018), we use as accumulation-rate input data the original accumulation-rates (acc) for the GISP2 site, as reconstructed in Cuffey and Clow (1997), but adapted to the GICC05 chronology (data availability: 100km: //doi.pangaea.de/10.1594/PANGAEA.888997). We use for the reconstruction all three

given accumulation-rate data-sets (called 50 km, 100 km or 200 km scenario) to analyse the impact of the three scenarios on the reconstructed temperature estimates.

### 2.3 Model Spin-Up

To avoid the influence of possible memory effects (influence of earlier firn-state conditions on later firn-states) on the model output of the reconstruction window, a temperature and accumulation-rate spin-up is needed in

order to bring the firn-model to a well-defined starting condition. For constructing the surface temperature spin-up we use the temperature reconstruction for the GISP2 site from Buizert et al. (2014) for the interval 10.05 kyr to 20 kyr b2k. The reconstruction is based on $\delta^{18}O_{ice}$-calibration and on $\delta^{15}N$-fitting using a dynamical firn-densification-model. We further extend the temperature spin-up to 35 kyr b2k by calibrating the GISP2 $\delta^{18}O_{ice}$ data (Grootes et al., 1993; Grootes and Stuiver, 1997; Meese et al., 1994; Steig et al., 1994; Stuiver et al., 1995;

data availability: Grootes and Stuiver, 1999) using the slope and intercept for the linear calibration given in Cuffey and Clow (1997). For the reconstruction window corresponding to the Holocene (0.02 kyr to 11.50 kyr b2k), we simply start with constant temperature using the last value of the spin-up section (10.05 kyr b2k, fig. 2, black line). Since we use three accumulation-rate scenarios (50 km, 100 km, 200 km) and a different firn-model as Buizert et al. (2014), it was necessary to adjust the model spin-up temperature in order match the decreasing

flank at the oldest part (9.5 kyr to 12.168 kyr b2k) of the gas-isotope data. The adjustment was done for all three used accumulation-rate scenarios separately. The constant offset of about 0.05 permil (or 4 K for the temperatures) between the modelled $\delta^{15}N$ using the unadjusted prior temperature and the adjusted temperatures potentially originates from two sources. First, the fact that the firn-model of Schwander et al. (1997) do not incorporate a convective zone and thereof a larger firn column is modelled. To model the gravitational

component of the isotope fractionation a higher absolute temperature is needed accelerating densification and leading to a reduced firn column (see also sect. 4.2). Second, the Schwander et al. (1997) firn-model do not model basal heat-flow which leads to a certain enrichment of the modelled isotopes compared to models which incorporate that quantity. Figure 2(a) shows the adjustment scheme for $\delta^{15}N$ data. The temperature spin-up was divided in different section indicated by the time markers A, B, C and D. The sections [start to A], [B to C], and

[D to end] were shifted with certain offsets to provide the best possible fit with the decreasing flank in the oldest part of the gas-isotope data. In-between A and B, and C and D the missing values were linearly interpolated. To find the three optimum offset parameter Nelder-Mead simplex minimisation was used (Lagarias et al., 1998). Figure 2(c) shows the modelled $\delta^{15}N$ values before (black) and after the adjustment (for different accumulation-rate scenarios: blue: 50 km, red: 100 km, yellow: 200 km) together with the measured $\delta^{15}N$ data (grey).

Figures 2(d) and 2(e) illustrates the differences between the model-outputs using the same spin-up adjustment (i.e. for 100 km acc-scenario) for all three accumulation-rate scenarios (d) and the outputs, where we conducted an individual adjustment for each scenario (e). It is shown that using the same spin-up for all three accumulation-rate scenarios leads to small but significant divergences between the model-outputs lasting for about 2 kyr ((d) vs. (e)).

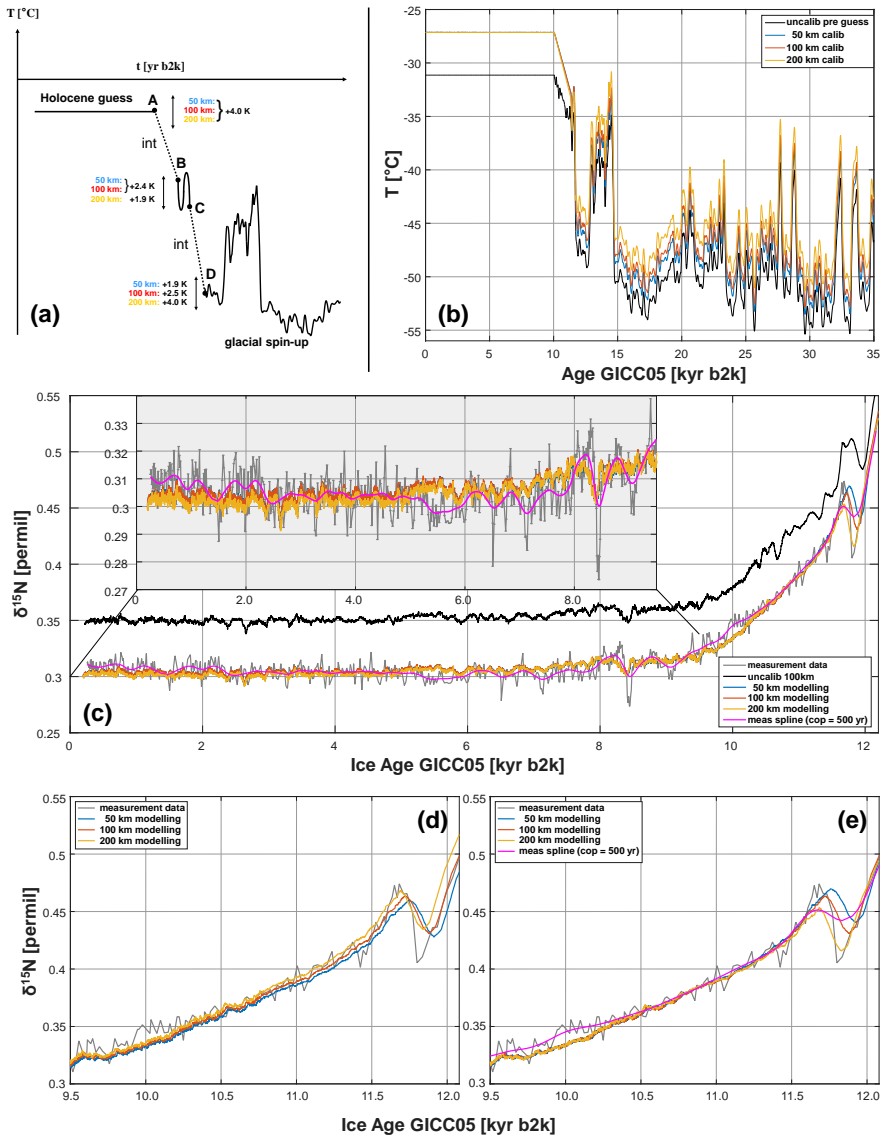

**Figure 2: (a):** Adjustment scheme of the prior inputs for different accumulation-rate scenarios (see text), the time of A-D are as follows: A = 10050 yr b2k, B = 11350 yr b2k, C = 11630 yr b2k, D = 11750 yr b2k; "int" refers to the regions of linear interpolation between the respective times after adjustment; the temperature offsets for the three parts of the time-series were found using Nelder-Mead Simplex minimisation (Lagarias et al., 1998). **(b):** Initial spin-up temperature (black line) with glacial section from Buizert et al. (2014) and Cuffey and Clow (1997), and adjusted input temperatures for different accumulation-rate scenarios (50 km: blue, 100 km: red, 200 km: yellow). **(c):** Raw (grey) and splined (purple) $\delta^{15}N$ measurement data (Kobashi et al., 2008b), and modelled $\delta^{15}N$ time-series using the initial spin-up (black), or the adjusted spin-up temperature inputs together with the respective accumulation-rates (50 km: blue, 100 km: red, 200 km: yellow). **(d):** Minimization window for the calibration: modelled $\delta^{15}N$ time-series using the 100 km calibration for all accumulation-rate scenarios showing that only the red curve (100 km) matches the introductory flank (11.8 kyr to 12.2 kyr) in the middle of variance of the measurement data sufficiently whereas the blue (50 km) and yellow (200 km) curve show divergences lasting for about 2 kyr (10 kyr to 12 kyr b2k). **(e):** Minimization window for the adjustment: modelled $\delta^{15}N$ time-series using the adjusted spin-up temperature input scenarios showing good agreement between the respective $\delta^{15}N$ scenarios.





### 3 Results and discussion: gas-isotope fitting

#### 3.1 Signal to noise analysis of isotope data

In order to evaluate the suitability of the three different inversion targets ($\delta^{15}$N, $\delta^{40}$Ar, $\delta^{15}$N$_{excess}$) on temperature reconstruction, we conducted a signal-to-noise analysis for all targets considering the given measurement

uncertainty as follows. First, we detrended the measured gas-isotope target data by subtracting the respective low-pass-filtered signals using a cut-off-period of 500 yr. Figure 3 shows the detrended high-frequency data (a-c). In a next step, we identify the local maxima and minima indicated by the red and blue triangles of the high-frequency isotope data (using the Matlab® "findpeaks" algorithm). Next, we define the high-frequency signals as the difference between successive local maxima and minima (fig. 3(d-f)) and compare the high-frequency

signals to the signal uncertainties calculated from the published measurement uncertainties (Kobashi et al., 2008b). As a signal is defined by at least two points the signal uncertainties are calculated using Gaussian-error-propagation. Kobashi et al. (2008) give different measurement uncertainties for different parts of the gas-isotope records and measurement campains. Consequently, we used for our calculations the minimum (red dotted line) and maximum (blue dotted line) uncertainties (tab.1) given in the Kobashi et al. publication. The analysis of the

high-frequency signals shows that for the minimum signal uncertainties 78% of the $\delta^{15}$N high-frequency signals have amplitudes higher than the uncertainty level (70% for the maximum uncertainty), 74% (or 36%) for $\delta^{40}$Ar, and only 52% (or 17%) for $\delta^{15}$N$_{excess}$, respectively. Assuming that the "true" uncertainty is in-between the given maximum and minimum uncertainties, and keeping in mind that the listed measurement uncertainty values are 1σ uncertainty, we argue that only $\delta^{15}$N is suitable as a robust reconstruction target in the high-frequency case

due to the high relative uncertainties compared to the signals of interest for the two other targets ($\delta^{40}$Ar, $\delta^{15}$N$_{excess}$). Figure 4 shows a detailed listing of the signal-to-noise ratios (SNRs) for all isotope species and for the minimum and maximum signal uncertainties, respectively. Here the signals are grouped between integer SNR values. It is clearly visible that for $\delta^{15}$N most signals have an SNR between one and two or above for the minimum as well as the maximum uncertainty. In contrast, for $\delta^{40}$Ar and especially for $\delta^{15}$N$_{excess}$ the dominant

fraction of signals have SNR values lower than the uncertainty values (SNR < 1), which makes it challenging to extract a robust temperature estimate from these targets.

For the longer term isotope trends (cop > 500 yr) it is more challenging to extract a comparable result (fig. 3(g-i)). Here we divided the in Kobashi et al. (2008b) stated measurement uncertainty by about 5.3 to account for the smoothing (mean data resolution = 17.8yr, sqrt(500yr/17.8) = 5.3). The comparison of the minimum or

maximum measurement uncertainties (red or blue error bars) indicates that all three gas-isotope quantities are suitable for reconstructing long-term temperature trends. This is particularly correct when only measurement uncertainty is considered as these uncertainties are in most of the cases lower than the amplitudes of the investigated features. However, $\delta^{15}$N is also the most suitable target for reconstruct long-term temperature trends due to its relative small uncertainty compared to $\delta^{40}$Ar and $\delta^{15}$N$_{excess}$. It has to be stressed that using $\delta^{15}$N$_{excess}$

only (as single target) for the reconstruction involves the danger of incorporating large drifts in the temperature solution due to the fact that the calculated temperature is solely dependent on the temporal integration of firn-temperature gradients. The elimination of the gravitational signal using $\delta^{15}$N$_{excess}$ as single target leads to a loss of information (LID) and to a less constraint temperature solution with reduced reproducibility (see 3.2.1 and 4.1).



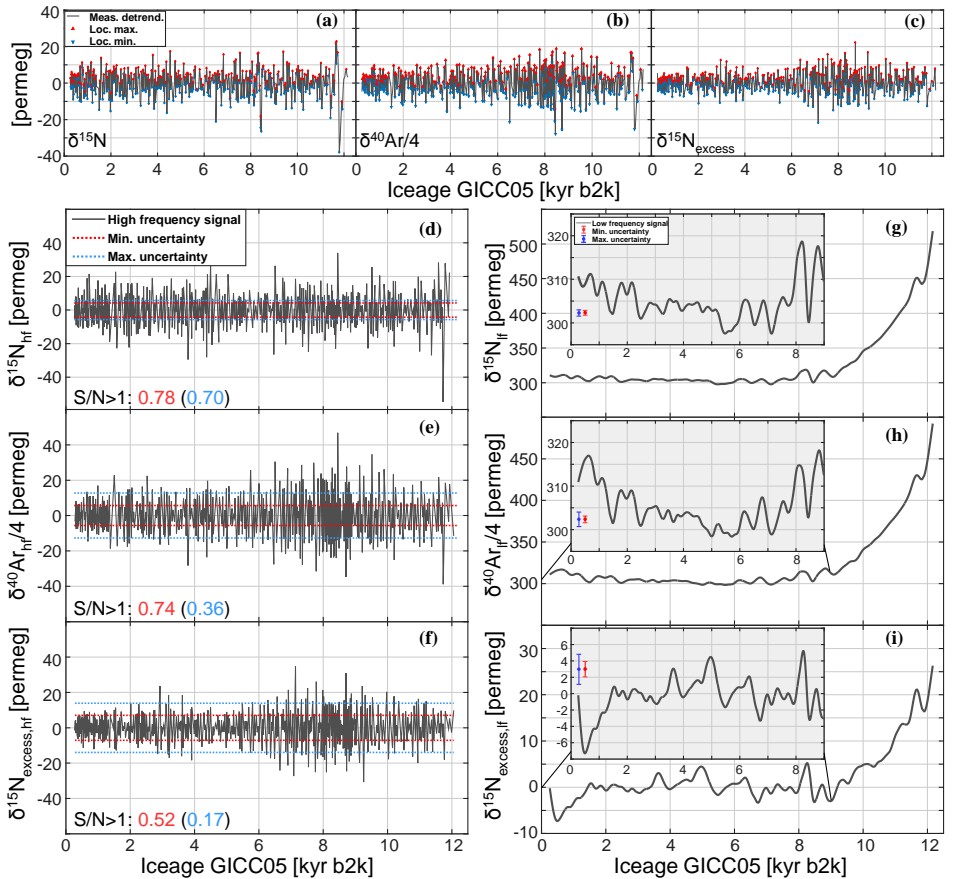

**Figure 3: Signal-to-noise analysis of the used target data (data from Kobashi et al., 2008b). (a-c): Detrended (cop = 500 yr) $\delta^{15}N$, $\delta^{40}Ar/4$ and $\delta^{15}N_{excess}$ high-frequency data. Triangles indicate local maxima (red) and local minima (blue) used to calculate the high-frequency signals. (d-f): $\delta^{15}N$, $\delta^{40}Ar/4$ and $\delta^{15}N_{excess}$ high-frequency signals (grey line) calculated from the differences of local minima and maxima of the detrended high-frequency data and minimum (red dotted line) and maximum (blue dotted line) signal uncertainties (see text). The red and blue numbers indicate the fraction of signals with amplitudes higher than the related uncertainties. (g-i): $\delta^{15}N$, $\delta^{40}Ar/4$ and $\delta^{15}N_{excess}$ low-frequency data (cop = 500 yr) and zoom-in for the last 9 kyr together with the maximum (blue) and minimum (red) measurement uncertainty (Kobashi et al., 2008b) divided by 5.3.**





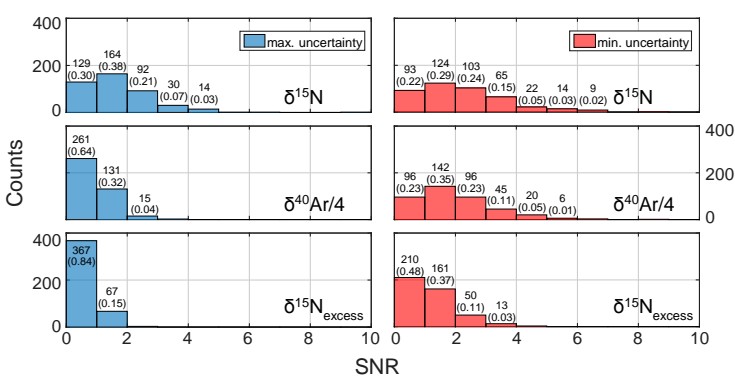

**Figure 4: Signal-to-noise-ratios (SNR) of the high-frequency signals for all reconstruction targets and for the maximum (left hand side) as well as the minimum (right hand side) signal uncertainties (see text). Values in the plots indicate the absolute (relative) number of signals with SNR in between the respective limits.**

### 3.2 Gas-isotope fitting results

In the following paragraphs we discuss the reconstructed temperature-solutions when fitting all different targets ($\delta^{15}$N, $\delta^{40}$Ar and $\delta^{15}$N$_{excess}$) in order to evaluate the reproducibility of the resulting solutions over ten fitting runs. Also we show the "goodness" of the fits when discussing the remaining final mismatches between modelled and measurement data. Finally we show the modelled gasage-iceage-differences ($\Delta$age) and lock-in-depth (LID) time-series for all fitting targets and a so called hybrid solutions. The hybrid solution is a combined approach created from the mean temperature solution of $\delta^{15}$N-fitting, low-pass filtered with a cut-off-period of 500 yr giving a long-term temperature trend. This long-term temperature trend is superimposed by adding high-frequency information calculated from $\delta^{15}$N$_{excess}$. The high-frequency temperatures are calculated by translating the mismatches of modelled – using the long-term temperature trend (from $\delta^{15}$N) – and measured $\delta^{15}$N$_{excess}$ data ($\Delta\delta^{15}$N$_{excess,hf}$) into temperature by using the empirical derived temperature sensitivities $\Omega_N$ of $\delta^{15}$N and $\Omega_{Ar}$ of $\delta^{40}$Ar as follows:

$$\Delta T_{hybrid,hf}(t) = \frac{\Delta\delta^{15}N_{excess,hf}(t)}{\left(\Omega_N(t) - \Omega_{Ar}(t)/4\right)} \tag{3}$$

The hybrid solution is used to imitate the temperature reconstruction by Kobashi et al. (2017) but with a different strategy. Kobashi et al. first fitted the temperature gradient over the diffusive firn-column $\Delta T_{firn}$ (eq. 4) calculated from $\delta^{15}$N$_{excess}$. Due to the high relative uncertainty of $\delta^{15}$N$_{excess}$ together with the yearly calculation of surface temperature from the modelled bottom temperature values using the $\Delta T_{firn}$-integration-method (Kobashi et al., 2008a, 2010) strong drifts in the reconstructed temperature can occur (see also sect. 4.1. and 4.5.). To overcome this issue Kobashi et al. forces the $\delta^{15}$N$_{excess}$ temperature to also fit the general trend of $\delta^{15}$N by allowing constant shifts in $\Delta T_{firn}$ in nine certain time windows of 1500 yr length. This correction reduces the goodness of the $\delta^{15}$N$_{excess}$-fit in some parts of the time-series and adds additional uncertainty to the reconstructed temperature. Additionally, the correction can change millennial scale trends, because changing the mean $\Delta T_{firn}$ in a 1500°yr window directly changes the temperature trend in this section. Also, allowing sharp shifts between the windows can create short term temperature signals (jumps with 50-200 yr durations) which could be misinterpreted as real temperature changes. As our method provides the possibility to know the long-term





T($\delta^{15}$N) trend, it is interesting to compare the hybrid temperature solution to the Kobashi et al. solution, and investigate the differences which should manly occure from the "window correction" method.

### 3.2.1 Reproducibility:

In order to evaluate the performance of the gas-isotope fitting algorithm coupled to the Schwander et al. (1997) firn-model, ten fitting runs were conducted for each gas-isotope species ($\delta^{15}$N, $\delta^{40}$Ar and $\delta^{15}$N$_{excess}$). From the ten solutions for each isotope a mean solution was calculated as the average of the ten temperature solutions. This mean-solution was run through the model giving the final modelled isotope solution. The gained data were analysed for the reproducibility between the ten runs (fig. 5) to give a measure for the possible spread in absolute temperature and isotope solutions for each isotope target. Also, we compare the final (mean) solution with the best-fit solution (out of ten) with regard to the remaining mismatches (fig.6) between the modelled and measured isotope data. Table 2 contains additional information for the reproducibility (rep) study as well as the goodness of the fits (fit) for the averaged and best-fit solutions. The top six plots of fig. 5 display the spread of the modelled isotope solutions as time-series and histograms. It is clearly visible that the spread of the isotope solutions between the ten runs is in the low permeg level for all targets with mean spreads over the whole reconstruction time-window (0.255 kyr to 12.168 kyr iceage b2k) of $1.68 \pm 1.14$ permeg for $\delta^{15}$N, $2.58 \pm 1.77$ permeg for $\delta^{40}$Ar/4, and $1.28 \pm 2.14$ permeg for $\delta^{15}$N$_{excess}$. The low variances between the modelled isotope solutions show the robust performance of our gas-isotope fitting algorithm. Similarly, the spread between the gained temperatures were analyse (bottom 6 plots of fig. 5). Using $\delta^{15}$N and $\delta^{40}$Ar as single targets lead to temperature solutions varying in a narrow band of $0.17 \pm 0.12$ K for T($\delta^{15}$N) and $0.26 \pm 0.18$ K for T($\delta^{40}$Ar). In contrast to the excellent reproducibility of T($\delta^{15}$N) and T($\delta^{40}$Ar), the fitting of $\delta^{15}$N$_{excess}$ generates a wide spread of possible temperature realizations which is in contrast to the robustness of the modelled isotope solutions. Whereas the variance between the isotope solutions is comparable for all three species the spread of T($\delta^{15}$N$_{excess}$) is with a value of $2.04 \pm 1.90$ K about 10 times higher as for T($\delta^{15}$N) or T($\delta^{40}$Ar). This is due to the removal of all information about the gravitational component in the isotope signals when calculating $\delta^{15}$N$_{excess}$ and therefore the information about the height of the firn column is lost. This leads to a significantly extended space of firn-states or absolute temperatures on which the fitting algorithm can work to yield very similar modelled isotope solutions. Besides the spread in absolute temperature, the relative temperature variations (deviation from the trend) gained by fitting $\delta^{15}$N$_{excess}$ are very much comparable to each other. However, the different densification backgrounds lead to differences in the time evolution of $\Delta$age of about 20 yr to 30 yr, and therefore in asynchronous temperature estimates (fig. 7). Due to this, fitting of $\delta^{15}$N$_{excess}$ as single target makes it challenging to determine the right timing of temperature changes on multi-decadal scale. Figure 7 shows the modelled $\Delta$age and LID for all fitting targets ($\delta^{15}$N, $\delta^{40}$Ar, $\delta^{15}$N$_{excess}$) and all 10 runs and the hybrid solution. Whereas the variance between the modelled $\Delta$age and LID time-series over 10 fitting runs is very small for $\delta^{15}$N and $\delta^{40}$Ar pointing to high reproducibility as discussed before, fitting of $\delta^{15}$N$_{excess}$ creates a variety of LID and $\Delta$age states in contrast to the comparable small misfits to the $\delta^{15}$N$_{excess}$ target data (see 3.2.2).



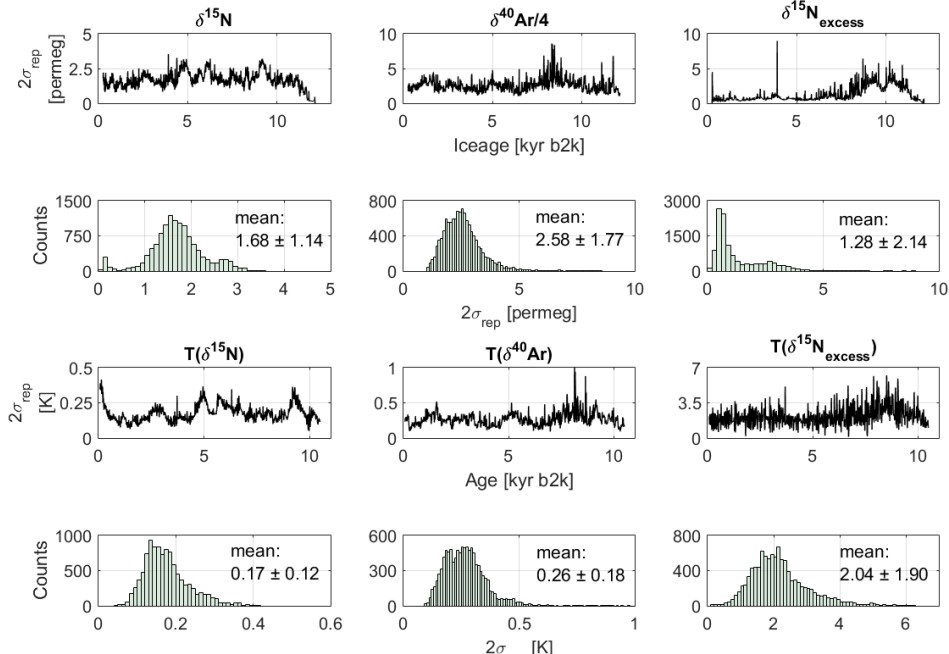

**Figure 5: Reproducibility between 10 runs for each target (first row: $\delta^{15}N$ and $T(\delta^{15}N)$ using the fitting algorithm coupled to the firn-model from Schwander et al. (1997); second row: $\delta^{40}Ar$ and $T(\delta^{40}Ar)$; third row: $\delta^{15}N_{excess}$ and $T(\delta^{15}N_{excess})$). Top 6 plots: reproducibility for the modelled isotopes per yr. Bottom 6 plots: reproducibility for the reconstructed temperatures per yr. Values are mean ± 2σ. See also table 2.**

**3.2.2 Final misfits:**

The top six plots of fig. 6 show the mismatch between the measured isotope data and the modelled isotope time-series using the respective averaged temperature solution as model input. The bottom 6 plots display the mismatches for the best-fit solution of the 10 runs for each isotope target, respectively. The numbers in the histograms are the 2σ values of all pointwise mismatches. The evaluation of the mismatches between the measured and modelled time-series gives a constrain on the uncertainty budget of the final temperature. It is obvious that for all targets the mismatches between the measured and modelled isotope data are at least comparable or below the analytic uncertainty (which is 1σ) of the measurement data (tab. 1). Using the average temperatures, we reach final mismatches (2σ) of 3.65 permeg for $\delta^{15}N$, 2.79 permeg for $\delta^{40}Ar/4$, and 5.43 permeg for $\delta^{15}N_{excess}$. An interesting fact is that for the $\delta^{15}N$- and $\delta^{40}Ar$-fits, the average temperature solution leads to a further decrease of the mismatch compared to the best fit out of the ten runs (4.5 % for $\delta^{15}N$, 18.9 % for $\delta^{40}Ar$). It seems that the averaging of the ten temperature solutions corrects some of the remaining (potential random distributed) mismatches. Obviously a larger number of runs (> 10) would slight improve the best solution. As discussed above, the averaging of the temperature solutions gained from $\delta^{15}N_{excess}$-fitting is problematic, due to the wider spread between the temperature solutions and thereof worse constrained Δage. Consequently, the averaging leads to an increase of the mismatches compared to the single fits. As shown in fig. 6, the averaging of the single $T(\delta^{15}N_{excess})$ solutions leads to more than a doubling (factor 2.55) of the mismatches compared to the best-fit solution.

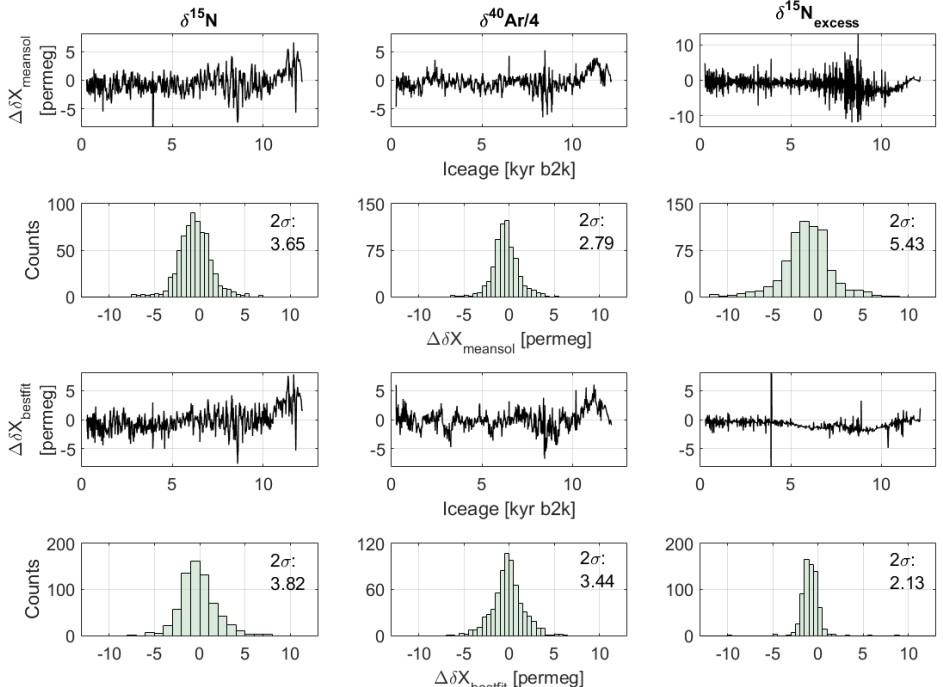

**Figure 6: Top 6 plots: Mismatches between the modelled and measured isotopes of the final (mean) solution. Bottom 6 plots: Mismatches between the modelled and measured isotopes of the best fit out of ten runs. Interesting is the decrease of the mismatches of the mean solution compared to the best fit for δ$^{15}$N and δ$^{40}$Ar. δ$^{15}$N$_{excess}$ shows a reverse behaviour due to the wider spread of "possible" temperature solutions (figure 5 and table 2).**



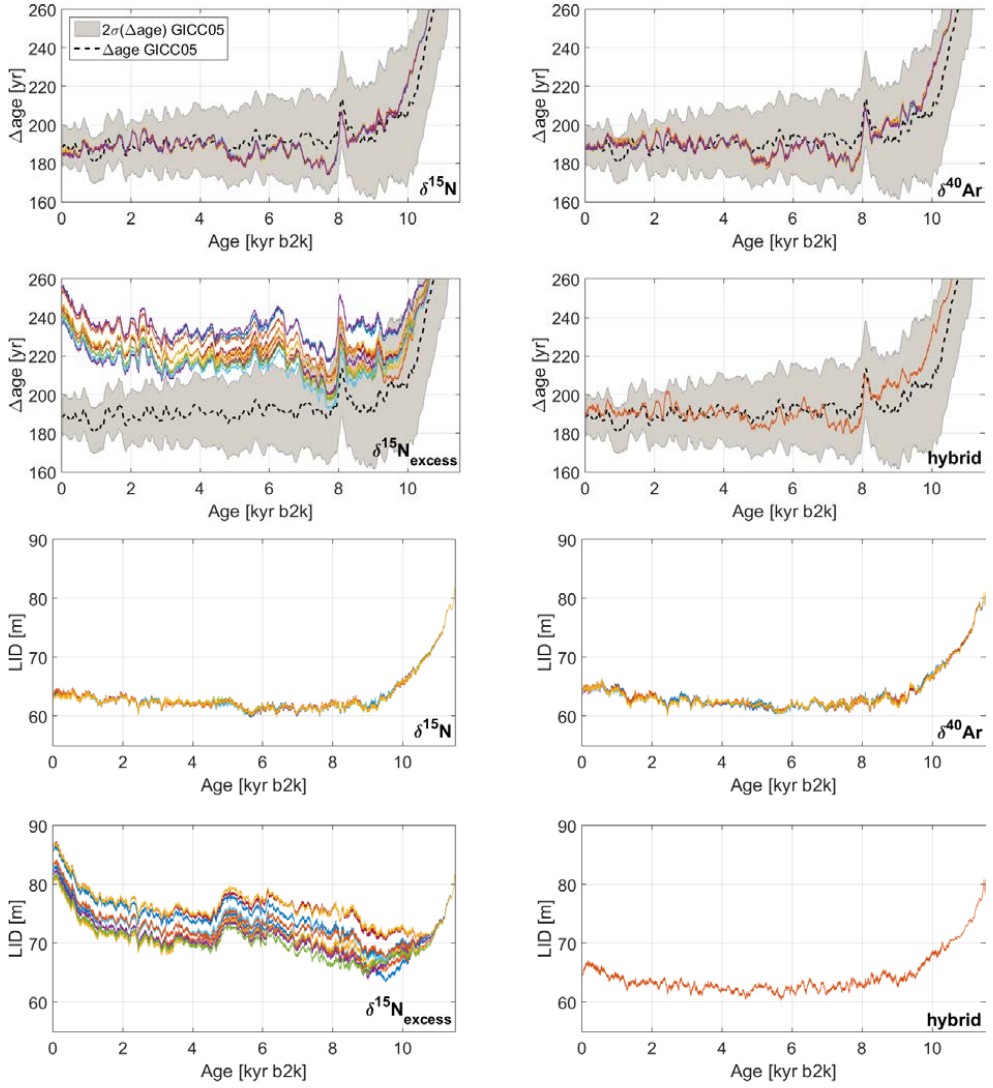

**Figure 7: Upper four plots: Modelled gasage-iceage-differences (Δage) for all isotope species (top left: δ$^{15}$N; top right: δ$^{40}$Ar; bottom left: δ$^{15}$N$_{excess}$; bottom right: hybrid) and the hybrid solution compared to the Δage for GISP2 from GICC05 (dashed black line). Notice the large spread for δ$^{15}$N$_{excess}$-fitting of about 20 yr to 30 yr.**
**Bottom four plots: Modelled lock-in-depth (LID) as a measure for the firn column height for all isotope species (top left: δ$^{15}$N; top right: δ$^{40}$Ar; bottom left: δ$^{15}$N$_{excess}$; bottom right: hybrid) and the hybrid solution. Notice the large spread for δ$^{15}$N$_{excess}$-fitting of about 10 m.**





| target | quantity | 50 km | 100 km | 200 km | information |
|---|---|---|---|---|---|
| $\delta^{15}N$ | mean($2\sigma_{miss}$) (best fit) [permeg] | 4.36 ± 0.40 (3.64) | 3.71 ± 0.38 (3.27) | 4.09 ± 0.19 (3.82) | fit |
| | $2\sigma_{missmatch}$ mean solution [permeg] | 3.83 | 3.19 | 3.65 | fit |
| | mean($2\sigma_{rep}$) [permeg] | 2.09 ± 1.64 | 2.16 ± 1.39 | 1.68 ± 1.14 | rep |
| $T(\delta^{15}N)$ | mean($2\sigma_{rep}$) [K] | 0.21 ± 0.17 | 0.22 ± 0.13 | 0.17 ± 0.12 | rep |
| $\delta^{40}Ar/4$ | mean($2\sigma_{miss}$) (best fit) [permeg] | | | 3.80 ± 0.24 (3.44) | fit |
| | $2\sigma_{miss}$ mean solution [permeg] | | | 2.79 | fit |
| | mean($2\sigma_{rep}$) [permeg] | | | 2.58 ± 1.77 | rep |
| $T(\delta^{40}Ar)$ | mean($2\sigma_{rep}$) [K] | | | 0.26 ± 0.18 | rep |
| $\delta^{15}N_{excess}$ | mean($2\sigma_{miss}$) (best fit) [permeg] | | | 2.93 ± 0.49 (2.13) | fit |
| | $2\sigma_{miss}$ mean solution [permeg] | | | 5.43* | fit |
| | mean($2\sigma_{rep}$) [permeg] | | | 1.28 ± 2.14 | rep |
| $T(\delta^{15}N_{excess})$ | mean($2\sigma_{rep}$) [K] | | | 2.04 ± 1.90* | rep |
| Hybrid ($\delta^{15}N$ lf, $\delta^{15}N_{excess}$ hf) | $2\sigma_{missmatch}(\delta^{15}N)$ $2\sigma_{missmatch}(\delta^{40}Ar)$ $2\sigma_{missmatch}(\delta^{15}N_{excess})$ [permeg] | | | 28.50 27.73 7.57 | fit |

**Table 2: Goodness of the fits (fit) and reproducibility (rep) using the firn-model from Schwander et al. (1997): For all gas-isotope targets 10 runs were conducted for the 200 km accumulation-rate scenario. Additionally 10 runs were conducted for each accumulation-rate scenario for $\delta^{15}N$ as target. mean($2\sigma_{missmatch}$) is the mean of the doubled standard deviation of the differences between modelled and measured data for all runs plus minus the $2\sigma$ deviation between the runs. mean($2\sigma_{missmatch}$) is a measure for the mean misfit between the modelled and the target data. mean($2\sigma_{solution}$) is the mean of the doubled standard deviations per age point over the 10 runs and a measure for the spread of the temperature solutions.**





### 3.2.3 Boundary effect

While running the gas-isotope fitting algorithm several times on the same target, we notice a boundary effect for the last 500 yr to 1 kyr to today when using the Schwander et al. as well as the Goujon et al. firn-model. Here different temperature solutions emerge while the rest of time-series is highly reproducible (fig. 8(c),(d) and Sect.

3.2.1). Figure 8 shows that issue for $\delta^{15}$N-fitting using the Goujon et al. firn-model. Fitting of $\delta^{15}$N leads to solutions with increasing (fig. 8(a): red lines), decreasing (fig. 8(a): blue lines) or even flat (fig. 8(a): green lines) millennial temperature trends in this time-window, but fitting the isotopes to the same precision (fig. 8(b),(c)). The reason for that is a cancellation between opposite trends of the thermal diffusion (fig. 8(e)) and the gravitational fractionation components (fig. 8(f)) of the modelled $\delta^{15}$N signals for those low magnitude signals

(about 20 permeg). For example, a long-term cooling trend will increase the diffusive firn column (and so the LID) due to decelerated densification of the firn, leading to an increase of the gravitational component of $\delta^{15}$N (fig. 8(f),(h): blue lines). On the same time the cooling leads to a decrease of the temperature gradient ($\Delta T_{firn}$) over the firn column (fig. 8(g)) which follows in first order the surface temperature trend. The decrease in $\Delta T_{firn}$ leads to a decrease of the thermal diffusion component (fig. 8(e)). The same happens for the warming

temperature trends but with opposite direction for LID(t) and $\Delta T_{firn}$(t). Only in the last 500 yr and due to the lack of data availability we experienced this boundary effect. This effect can be understood by the influence of data on the past values. The fitting algorithm works from past into future direction. If a "wrong" temperature trend would be created leading to the same $\delta^{15}$N signal as for the "right" temperature trend, there will be a certain point in time in future direction where the modelled isotope signals will drift away from the measured isotope targets

leaving the range of the cancellation. Unfortunately, at the boundary of the gas-isotope record there is no further data available to stabilize the temperature solutions. Using this explanation for testing the reproducibility (in 3.2.1), we extended the measurement data for 1 kyr into the future by adding constant isotope values calculated from the mean of the isotope data over the recent 1 kyr. This extension leads to stable temperature solution even for the last 1 kyr and force the algorithm to produce the flat temperature solution in this time-window.

Unfortunately, it cannot be determined in that way which temperature trend for the last 1 kyr is the most realistic case by fitting the isotope data as single targets. To overcome this problem an additional constraint is needed. For that issue we use an analogue to Kobashi et al. (2017) (see supplement fig. S4 there), namely using the measured borehole temperature profile for the GISP2 site (Alley and Koci, 1990; Clow et al., 1996). The firn-model from Goujon et al. (2003) provides the possibility to model the temperature profile through the ice-sheet.

As the measurement target data ($\delta^{15}$N) starts from 255 yr b2k (GICC05 ice ages) which refers to 70 yr b2k (gas ages, modelled from T($\delta^{15}$N)) we had to extend the temperature from 70 yr b2k to 7 yr b2k, the time when the temperature profile was measured. For this extension we use Greenland Summit temperatures from Kobashi et al. (2017) estimated from correlating coastal and Summit instrumental air temperatures. Figure 8(i) shows the modelled temperature profile for all 10 temperature scenarios at the boundary of the gas-isotope record. For

correctly model the measured temperature profile we had to shift each of the 10 temperature estimates by a constant offset of -1 K for the entire Holocene (70 yr b2k to 11.65 kyr b2k) changing the absolute temperature but not the anomalies. The reason for this necessity remains unknown yet. It is clearly visible that only those scenarios with a cooling trend (blue lines) leading to an acceptable shape when compared to the measured borehole profile. Additionally it is possible to use $\delta^{40}$Ar data as second constrain (fig. 8(j)). Comparing the

measured and modelled $\delta^{40}$Ar data also favours the temperature estimates showing the cooling trend because these solutions are leading to modelled $\delta^{40}$Ar with the smallest mismatch to the measured data.

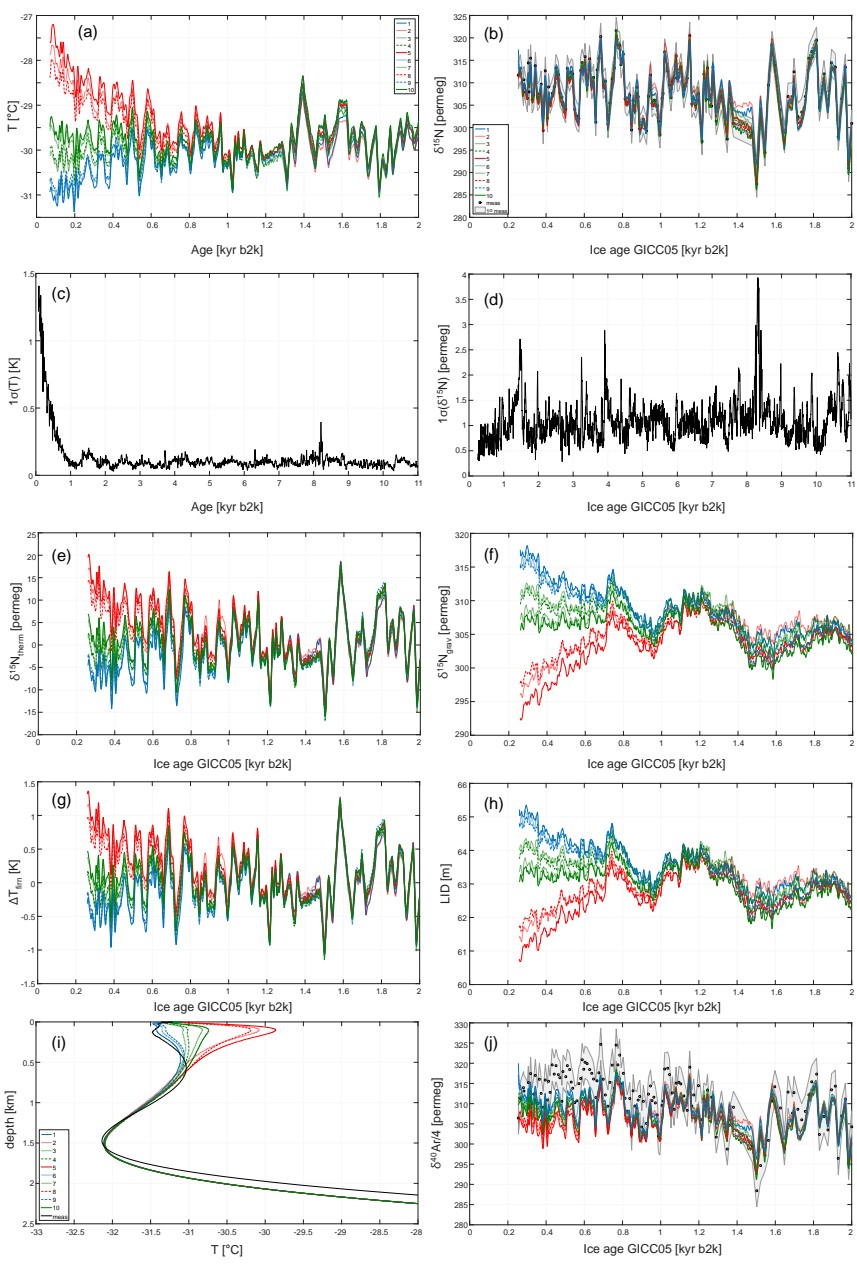

Figure 8: (a): Deviation of reconstructed temperatures on the boundary of the gas-isotope record between 10 runs of the fitting algorithm. (b): Measured δ15N record (dots) with minimal 1σ uncertainty (grey area) and δ15N-fits modelled using the temperatures from (a). (c) Standard deviation (spread) between the 10 temperatures from (a) showing a decreasing reproducibility on the boundary for the last 0.5-1.0 kyr to today. (d): Standard deviation (spread) between the 10 δ15N-fits does not show decreasing reproducibility on the boundary for the last 500 yr to 1 kyr to today. (e): Thermal fractionation component of δ15N modelled using the reconstructed temperatures of (a). (f) Component of δ15N due to gravitational settling modelled using the reconstructed temperatures of (a). (g) Firn temperature gradient modelled using the reconstructed temperatures of (a). (h) Lock-in-depth modelled using the reconstructed temperatures of (a). (i) Temperature profile through the ice-sheet modelled using the reconstructed temperatures of (a). (j): Measured δ40Ar/4 record (dots) with minimal 1σ uncertainty (grey area) and δ40Ar/4 modelled using the temperatures from (a).





### 3.2.4 Influence of different accumulation-rate estimates

To investigate the contribution of the uncertainty in the used accumulation-rate data, we use all three available accumulation-rate estimates for the GISP2 site (see sect. 2.2 and 2.3) to reconstruct temperature on the base of $\delta^{15}$N-fitting. Figure 9 shows the results of this analysis. Figure 9(a) shows the accumulation-rate raw data. Here the differences between the three scenarios are hardly visible. In fig. 9(b) the deviation of each single scenario to the average of all three scenarios is shown. It is visible that this deviation can be up to more than ±10% in the early-Holocene and decreases over time. Subplot (c) of fig.9 shows the maximum spread in modelled Δage emerging when using the three accumulation-rate scenarios for the temperature reconstruction for the whole input time-window. Subplot (d) shows the same for the Holocene part. It is clearly visible that the differences in Δage between the three scenarios have a small influence on the modelled Δage during the Holocene. Starting with a maximum difference of about 30 yr in the early-Holocene, the Δage difference decreasing until 5 kyr b2k following the decrease in the difference between the accumulation-rate data. From the mid-Holocene until today the modelled Δage difference becomes less than 5 yr. The same is true for the spread in LID which is shown in subplot (e). Here also the effect of the adjustment is visible forcing the same firn-state for all accumulation-rate scenarios at the beginning of the reconstruction window (see sect. 2.3). Without the pre-calibration it would be impossible to fit the $\delta^{15}$N data below the measurement uncertainty in the early-Holocene. Subplot (f) shows the long-term fraction (cop > 500 yr) of the reconstructed temperatures. Whereas the reconstructed temperatures when using the 50 km and 100 km accumulation-rate scenarios leading mainly to the same temperature trend, the reconstruction using the 200 km scenario showing a slightly larger cooling over the Holocene starting from about 7.5 kyr b2k. This is exactly the point in time where the decrease in accumulation-rate for the 200 km scenario compared to the averaged scenario starts to accelerate (see subplot (b)). As we see not an equal (but opposite) behaviour for the 50 km scenario we may have found a non-linear response between temperature and accumulation-rate change. From about 6 kyr to today this faster cooling starting at 7.5 kyr levels out and leads to a constant offset of about 0.3 K between the 200 km solution and the two others, which is nearly 15% of the whole cooling trend of about 2 K referred to the warmest part of the reconstructed temperatures at around 7.8 kyr b2k. Besides this, the shapes and amplitudes of the faster signals of the long-term fractions are highly comparable. Subplots (g) and (h) contains a zoom-in for the high-frequency temperature fractions calculated by subtracting the long-term fractions of subplot F from the reconstructed temperatures. In subplot G we clearly see the larger deviation of Δage between the scenarios leading to a slightly asynchronous behaviour of the short term temperature fractions. But the shapes and amplitudes of the signals are more or less independent from the deviations between the accumulation-rate scenarios. In subplot (h) the effect of the decreasing deviation between the accumulation-rate scenarios and thereof between the modelled Δage scenarios is shown. Here we find no differences between the short term fractions of the reconstructed temperatures. To sum up, the deviation between the three different accumulation-rate scenarios do not have a major impact on the reconstructed temperature anomalies. In first order the differences between the accumulation-rates leads to slightly different modelled Δage in the early-Holocene and to a 0.3 K larger cooling for the higher accumulation-rate scenario compared to the two other ones.



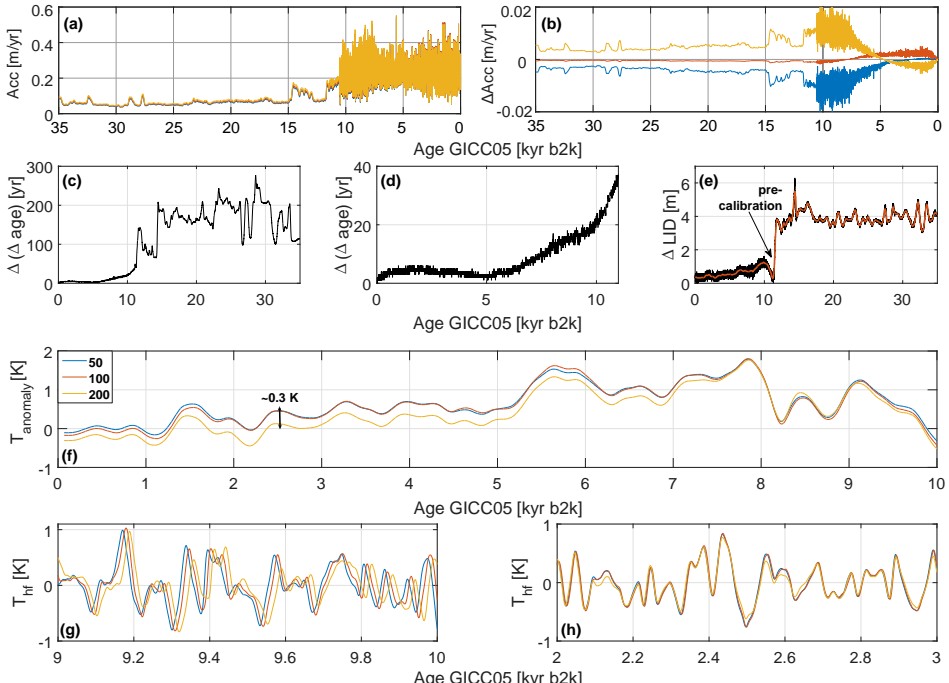

**Figure 9:** Influence of different accumulation-rate scenarios ((a): blue: 50 km, red: 100 km, yellow: 200 km) on the reconstruction. Please note, for plots (a) and (b) the x axis is inverted. B: Deviation of each single scenario to the average of all three scenarios. (c): Maximum spread in modelled Δage using the three different accumulation-rate scenarios for the whole input (Holocene and spin-up). (d): Zoom-in for (a) for the Holocene part. (e): Maximum spread in the modelled Lock-In-Depth (LID), the pre-calibration leads to a convergence of the LID and therefor to the same gravitational background for the isotope signals. (f): Long-term temperature trend $T_{anomaly}$ (low-pass: > 500 yr) of the reconstructed temperatures modelled using the three accumulation-rate scenarios. (g): Short-term temperature signals $T_{hf}$ (high-pass: < 500yr) showing asynchrony in the early-Holocene, as a result of the spread in Δage. (h): Short-term temperature signals $T_{hf}$ (high-pass: < 500yr) showing synchrony from the middle to late-Holocene, as a result of the decreasing spread in Δage and the decreasing difference between the accumulation-rates.

## 4 Results and discussion: temperature reconstruction

### 4.1 Solution comparison

(i) Comparison of $\delta^{15}$N and $\delta^{40}$Ar reconstruction

Figure 10 contains the results of temperature reconstruction when using $\delta^{15}$N or $\delta^{40}$Ar as fitting-targets together with the Schwander et al. (1997) firn-model. In fig. 10(a) the reconstructed temperatures and the differences (black curve) between $T(\delta^{15}$N) and $T(\delta^{40}$Ar) are shown. For visibility reasons the reconstructed temperatures were smoothed with a 100 yr cut-off-period (thick lines). It is obvious that the general trends between both reconstructions are very similar. Also the shapes of many of the shorter term temperature features are in a good agreement, but $T(\delta^{40}$Ar) points to higher amplitudes of these anomalies. The mean offset between $T(\delta^{15}$N) and $T(\delta^{40}$Ar) over the whole time-series is 0.28 K and the standard deviation (2σ) of the differences is 1.00 K for the maximum resolution case (mean resolution of the isotope data is 17.8 yr). In the early to mid (6.4 kyr-11.5 kyr) and late-Holocene (0.07 kyr-1.3 kyr b2k) the $\delta^{15}$N reconstruction leads to a slightly higher absolute temperature compared to $T(\delta^{40}$Ar) with mean-offsets of about 0.46 K for the early to mid-Holocene and 0.61 K for the late-Holocene, whereas the rest of the temperature time-series is showing a similar trend (1.3 kyr-6.4 kyr b2k, mean



offset: 0.06 K). For a further comparison of both reconstructions we correlate $T(\delta^{15}N)$ and $T(\delta^{40}Ar)$ after low-pass filtering (cop = 50 yr, tab.4) and band-pass filtering in three periodic-time bands: multi-decadal (band: 50 yr-200 yr, tab.5), multi-centennial (band: 200 yr-1000 yr, tab.6), and multi-millennial (1000 yr-4000 yr, tab.7). The correlations were calculated in the time-window 0.5-11.5 kyr b2k to account for the boundary effect

(sect. 3.2.3). Statistical significance (p-value) was calculated using the approach of Ebisuzaki (1997). The same analysis was conducted for all other targets and for the solutions gained by both firn-models. We find high correlations for the low-pass filtered data (r = 0.96, p < 0.01) and the multi-millennial band (r = 0.94, p < 0.01), which was expected due to the high agreement in the long-term trends between both temperatures. The multi-centennial band shows a lower but still high correlation (r = 0.87, p < 0.01). As the slightly different $\Delta$age

regimes can alter the correlation for the multi-decadal band, we conducted sample-cross-correlation (*xcf*, Box et al., 1994) to find the time-lag between $T(\delta^{15}N)$ and $T(\delta^{40}Ar)$ showing maximum correlation. In the multi-decadal band (tab.5) the correlation between $T(\delta^{15}N)$ and $T(\delta^{40}Ar)$ is weak (r = 0.69, lag = -4 yr or r = 0.67 for lag = 0 yr) but still significant (p < 0.01), and equals the correlation between the measured isotopes ($\delta^{15}N$ and $\delta^{40}Ar$) in the same band with r = 0.68 (p < 0.01). This result is not surprising due to two facts: (i) The high-frequency fraction

of the reconstructed temperatures is directly calculated from the high-frequency fraction of the respective isotope targets (Döring and Leuenberger, 2018) and (ii) the accumulation-rate input has only a minor effect on the reconstructed temperatures in multi-decadal band. Additionally, the slope between the measured isotope data ($\delta^{15}N(y)$ and $\delta^{40}Ar/4(x)$) for the multi-decadal case was calculated with a value of m = 0.89 ± 0.05 using geometric-mean-regression (Leng et al., 2007) to account for the uncertainty in the isotope data. This result

cannot be explained by thermal fractionation. The ratio between the thermal fractionation sensitivities (for Holocene conditions) of $\delta^{15}N$ to $\delta^{40}Ar/4$ ($\Omega_N/[\Omega_{Ar}/4]$) is about 1.46. The slope of 0.89 is pointing to systematic too high $\delta^{40}Ar$ values (see below), may be caused either from higher noise of the $\delta^{40}Ar$ data or potential gas-loss (see next section). It was not possible to estimate the exact reason for that (see also sect. 4.2.). Figure 10(b) shows the comparison between the modelled LIDs when fitting $\delta^{15}N$ (blue line) or $\delta^{40}Ar$ (purple line). Generally

both LID estimates are in good agreement. The differences between them vary in a narrow band of -2 m to 1 m (which equals less than 3.5% of relative disagreement) driven by the temperature differences between the reconstructions. The slightly lower temperature of $T(\delta^{40}Ar)$ in the early to mid (6.4 kyr-11.5 kyr) and in the late-Holocene (0.07 kyr-1.3 kyr b2k) compared to $T(\delta^{15}N)$ leads to a larger LID in that time due to decelerated densification with lower temperatures. Figure 10(c) displays the modelled temperature gradient over the

diffusive firn column ($\Delta T_{firn}$) for the $\delta^{15}N$ (blue line) and $\delta^{40}Ar$ (purple line) fit. Additionally $\Delta T_{firn,meas}$ calculated directly from the measured isotope data (dotted line, meas) together with its maximum ($1\sigma_{max}$) and minimum ($1\sigma_{min}$) uncertainty. $\Delta T_{firn,meas}$ was calculated analogous to Kobashi et al. (2010, 2011, 2017) from $\delta^{15}N_{excess}$ according to:

$$\Delta T_{firn,meas} = \frac{\delta^{15}N_{excess}}{\left(\Omega_N - \frac{\Omega_{Ar}}{4}\right)} = \frac{\delta^{15}N - \frac{\delta^{40}Ar}{4}}{\left(\Omega_N - \frac{\Omega_{Ar}}{4}\right)} \tag{4}$$

The uncertainties ($1\sigma_{max}$, $1\sigma_{min}$) of $\Delta T_{firn,meas}$ were calculated using Gaussian-error-propagation on eq. 4 together with the uncertainties of $\delta^{15}N$ and $\delta^{40}Ar$ as stated in tab. 1. The comparison of modelled $\Delta T_{firn}$ of the $\delta^{15}N$- and $\delta^{40}Ar$-fits shows a high agreement in the general trends and also in the shapes of the shorter term features. The differences between them are less than ±1 K in most of the case. If compared to the $\Delta T_{firn,meas}$ we find a good agreement of the general trend in the late to mid-Holocene (1.3 kyr-6.4 kyr b2k) which is the part in time where

also the trends in $T(\delta^{15}N)$ and $T(\delta^{40}Ar)$ showing the smallest offset. In the early to mid-Holocene (6.4 kyr-



11.5 kyr) $\Delta T_{firn}$ modelled from $\delta^{15}N$ and $\delta^{40}Ar$ significantly exceeds $\Delta T_{firn,meas}$ which can be a sign for systematic too high $\delta^{40}Ar$ in this section, reducing $\delta^{15}N_{excess}$ and $\Delta T_{firn,meas.}$ The same is partly true for the late-Holocene (0.07 kyr-1.3 kyr b2k), but it has to be mentioned that the outcomes of the flat temperature realisations (sect. 3.2.3.) are shown here, which raise the mismatched between the modelled and measured $\Delta T_{firn}$ especially in the last 500 yr to today. However, the analysis of the $\delta^{15}N_{excess}$-fits (see next section) also pointing to too low $\Delta T_{firn,meas}$ in this time period. Comparing the amplitudes of the faster signals of the measured and modelled $\Delta T_{firn}$ shows that the modelled signals underestimates the amplitudes of $\Delta T_{firn,meas}$ which leads to the assumption that $\Delta T_{firn,meas}$ and therefore $\delta^{15}N_{excess}$ is potentially more driven by noise in the isotope data than temperature.



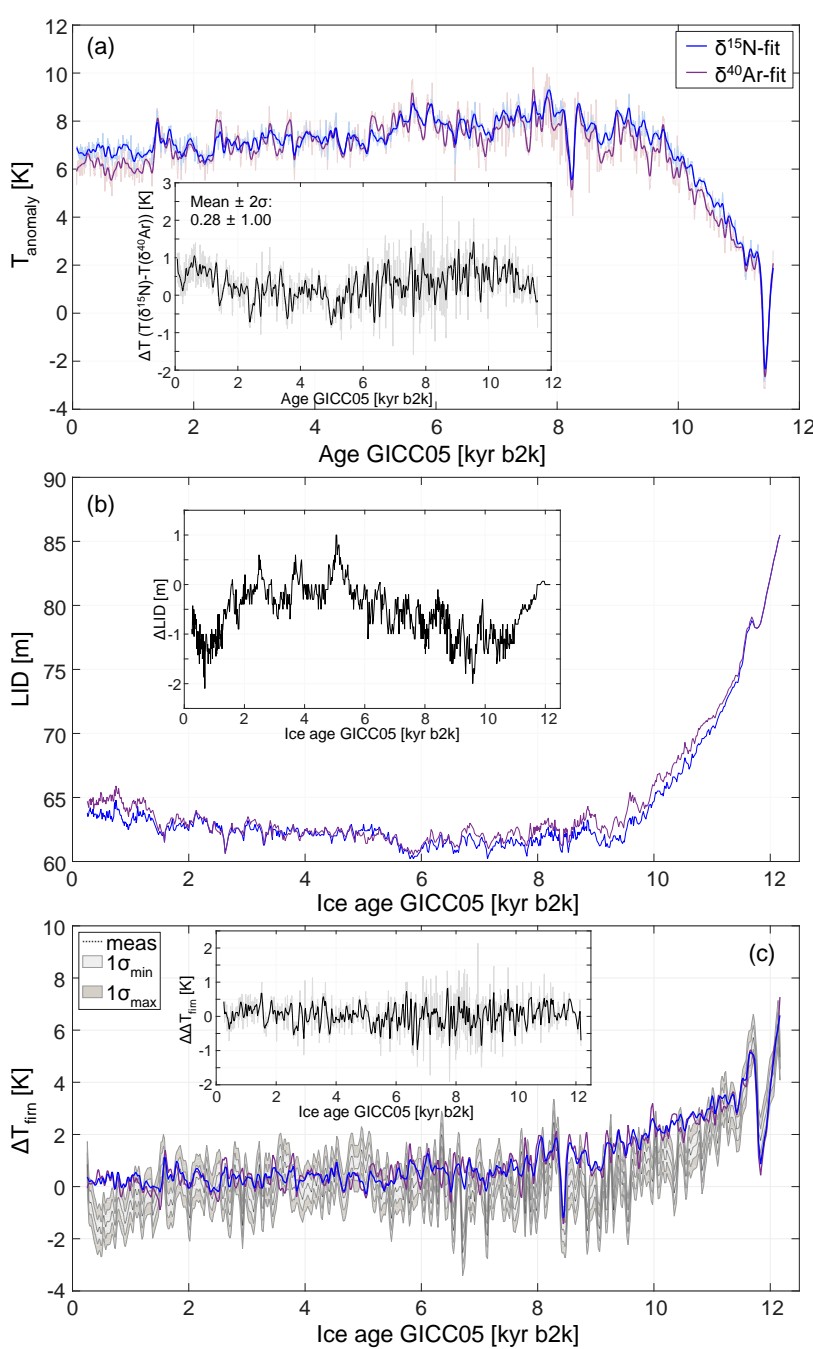

**Figure 10: Comparison between temperature anomalies (a), LID (b) and $\Delta T_{firn}$ (c) reconstructed from $\delta^{15}N$ (blue) and $\delta^{40}Ar$ (purple) using the model from Schwander et al. (1997). Black curves in the subplots show the differences between the $\delta^{15}N$- and $\delta^{40}Ar$-fits for the given quantities. In subplot (c) the modelled data is shown together with measured data (meas, dotted line) from Kobashi et al. (2008) with minimal and maximal 1σ uncertainty calculated from uncertainties given in tab. 1 with eq. (4).**





(ii) Comparison of $\delta^{15}N$ and $\delta^{15}N_{excess}$ reconstruction

The comparison of the results of $\delta^{15}N$ (blue) and $\delta^{15}N_{excess}$ (red, best fit) fitting are shown in Fig. 11, where subplot (a) displays the reconstructed temperature estimates, subplot (b) the modelled LID and (c) the modelled and measured $\Delta T_{firn}$. It is clearly visible that precise fitting of $\delta^{15}N_{excess}$ and therefore $\Delta T_{firn}$ (see (c)) results in a distinct temperature regime compared to $T(\delta^{15}N)$ and $T(\delta^{40}Ar)$. In the early-Holocene (9 kyr-11.5 kyr) the fitting of $\delta^{15}N_{excess}$ leads to a flat temperature with nearly no trend. This is also visible for the mean solution and the estimate using the Goujon et al. firn-model (see fig. 12(d)). This is not only in disagreement with $T(\delta^{15}N)$ or $T(\delta^{40}Ar)$ but also with the reconstructions from Kobashi et al. (2017) and Buizert et al. (2018) for the GISP2 site (see 4.1.3 and 4.1.4). The flat temperatures in this section when precisely fitting $\delta^{15}N_{excess}$ is pointing to too low $\delta^{15}N_{excess}$ and therefore $\Delta T_{firn}$, which could be driven by too high $\delta^{40}Ar$ in that section. In the late-Holocene $T(\delta^{15}N_{excess})$ shows a large cooling of about -3.6 K/kyr (0.16 kyr-1.25 kyr) which is highly unrealistic, pointing also to too low $\delta^{15}N_{excess}$ and $\Delta T_{firn}$. Also the $\delta^{15}N_{excess}$-fit of Kobashi et al. (2017) showing large disagreement between their modelled $\delta^{15}N_{excess}$ and the measured ones during this section (see supplementary information, fig.S3), which means that the quality of the $\delta^{15}N_{excess}$-fit has to be reduced significantly to extract a potentially meaningful temperature estimate. It has to be mentioned that $\delta^{40}Ar$ data are influenced by post-coring gas-loss (Kobashi et al., 2008b, 2010, 2011), which makes a correction based on firn-modelling (Kobashi et al., 2008a) and on $\delta Ar/N_2$ (a tracer for potential gas-loss; Kobashi et al., 2010, 2017) mandatory, leading to larger $\delta^{40}Ar$ values compared to the uncorrected $\delta^{40}Ar$. As we have used the corrected $\delta^{40}Ar$ to calculate $\delta^{15}N_{excess}$ and $\Delta T_{firn}$ we have to argue that the presently available correction (Kobashi et al., 2015b, 2017) is non-sufficient, especially for the late and early-Holocene data. This result is somehow surprising as it was argued in Buizert et al. (2018) that the influence of possible gas-loss on $\delta^{40}Ar$ is most severe within the bubble-clathrate transition zone (about 800 m to 1500 m depth of the GISP2 core, equals 3.8-9.3 kyr BP ice age, see supplement p. X-2 in Buizert et al. (2018)). In the following we compare two of the major mid-Holocene cooling trends in $T(\delta^{15}N_{excess})$ with the simultaneous trends in $T(\delta^{15}N)$. In the time range 2 kyr to 4.8 kyr b2k, where the trends in $T(\delta^{15}N)$ and $T(\delta^{40}Ar)$ are highly comparable, $T(\delta^{15}N_{excess})$ shows a cooling rate of -0.52 K/kyr. The cooling trend of $T(\delta^{15}N)$ in the same time range is -0.18 K/kyr and therefore about 3 times smaller. In the second time range from 6.3 kyr to 8.1 kyr, where $T(\delta^{15}N)$ and $T(\delta^{40}Ar)$ show a significant offset, the cooling in $T(\delta^{15}N_{excess})$ exceeds the cooling in $T(\delta^{15}N)$ by a factor of about 2.5 (-1.58 K/kyr for $T(\delta^{15}N_{excess})$, -0.62 K/kyr for $T(\delta^{15}N)$). Also these results could be explained by too low $\delta^{15}N_{excess}$ and $\Delta T_{firn}$. Next, we compare the correlations of $T(\delta^{15}N_{excess})$ with $T(\delta^{15}N)$ and $T(\delta^{40}Ar)$ in the same periodic-time bands as it was done in the previous section. In all periodic-time bands the correlation of $T(\delta^{15}N_{excess})$ with $T(\delta^{15}N)$ as well as $T(\delta^{15}N_{excess})$ with $T(\delta^{40}Ar)$ is weak. For the low-pass filtered data (tab.4) the correlations between $T(\delta^{15}N_{excess})$ and $T(\delta^{15}N)$ as well as $T(\delta^{15}N_{excess})$ with $T(\delta^{40}Ar)$ are not significant ($p > 0.1$ for all cases) and small ($|r| < 0.35$ for all cases). The highest correlation between $T(\delta^{15}N_{excess})$ and $T(\delta^{15}N)$ was found for the multi-millennial time-band (tab.7) with $r = 0.61$ ($p < 0.01$) for the best fit $T(\delta^{15}N_{excess})$ solution and $r = 0.68$ ($p < 0.01$) for the averaged $T(\delta^{15}N_{excess})$ solution. The correlation with $T(\delta^{40}Ar)$ is even weaker ($r = 0.48$, best fit; $r = 0.54$, mean solution; $p < 0.01$ for both). For the multi-centennial band (tab.6) the correlations are further reduced. For corr($T(\delta^{15}N_{excess})$, $T(\delta^{15}N)$) we find $r = 0.54$ ($p < 0.01$) or $r = 0.56$ ($p < 0.01$) for the best-fit solution or the mean solution and $r = 0.20$ ($p = 0.03$) and $r = 0.19$ ($p = 0.05$) for corr($T(\delta^{15}N_{excess})$, $T(\delta^{40}Ar)$). In the multi-decadal time-band (tab.5) we find a significant ($p < 0.01$ for best fit $T(\delta^{15}N_{excess})$, $p = 0.07$ for averaged $T(\delta^{15}N_{excess})$) but also weak negative correlation between $T(\delta^{15}N_{excess})$ and $T(\delta^{40}Ar)$ with $r = -0.41$ and $r = -0.36$ for the best fit and the mean $T(\delta^{15}N_{excess})$ solution, respectively. The result





for the multi-decadal time band points to the fact that the multi-decadal oscillations in $T(\delta^{15}N_{excess})$ are mainly driven by $\delta^{40}Ar$ with less influence from $\delta^{15}N$ due to higher variability of $\delta^{40}Ar/4$ compared to $\delta^{15}N$ (see previous section).

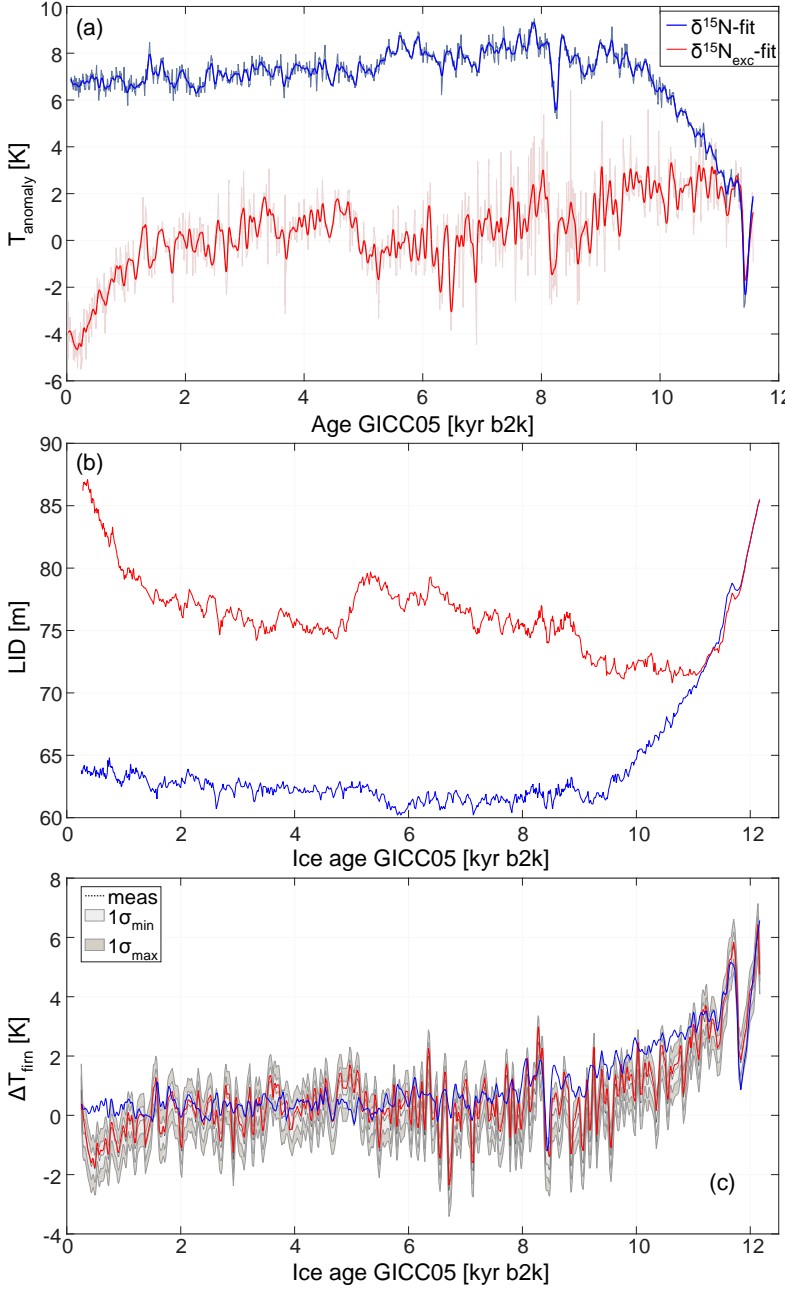

5    **Figure 11: Comparison between temperature anomalies (a), LID (b) and $\Delta T_{firn}$ (c) reconstructed from $\delta^{15}N$ (blue) and $\delta^{15}N_{excess}$ (red) using the model from Schwander et al. (1997). In subplot (c) the modelled data is shown together with measured data (meas, dotted line) from Kobashi et al. (2008b) with minimal and maximal 1σ uncertainty calculated from uncertainties given in tab.1 with eq. (4).**





(iii) Comparison of misfits among the reconstructions

The histograms on the right-hand-side of fig. 12 show the mean mismatches for all gas-isotope quantities when fitting a single isotope target using the Schwander et al. (1997) (black bars) or the Goujon et al. (2003) (blue bars) firn-model. The histogram on top shows that issue for the $\delta^{15}N$ reconstruction. E.g. precise fitting of $\delta^{15}N$ (where 96% of the mismatches (2σ) are smaller than 3.7 permeg using Schwander et al. (1997) firn-model) leads to insufficient fits for the other gas-isotope targets with mismatches (2σ values) of 11.3 permeg for $\delta^{40}Ar/4$ and 11.1 permeg for $\delta^{15}N_{excess}$, keeping in mind that the data uncertainty (1σ) of the measured gas isotope data is 3.0 to 4.0 permeg for $\delta^{15}N$, 4.0 to 9.0 permeg for $\delta^{40}Ar/4$ and 5.0 to 9.8 permeg for $\delta^{15}N_{excess}$. In other words, precise fitting of $\delta^{15}N$ do not automatically leads to accurate fits for $\delta^{40}Ar$ and $\delta^{15}N_{excess}$. The same is true for all other single fits and when using both firn-models. It is not possible to find a temperature estimate leading to modelled isotope regimes that provide a sufficient agreement for all isotopic targets together. This finding is pointing to the issues that the GISP2 gas-isotope data suffers from fractionations which are not captured by the used firn-models (e.g. due to post-coring gas-loss). As $\delta^{15}N$ is easier to measure due to the higher abundance of nitrogen in air and less susceptible to post-coring gas-loss (Huber et al., 2006a), we argue that $T(\delta^{15}N)$ provides the most robust temperature estimate compared to $T(\delta^{40}Ar)$ and especially $T(\delta^{15}N_{excess})$.

## 4.2 Bandpass analysis of isotope data

As discussed in sect. 4.1, the linear dependency (slope) between $\delta^{15}N$ and $\delta^{40}Ar$ deviates from the theoretical expectation for the multi-decadal band. Here we investigate this dependency for the same periodic-time bands as discussed in the previous sections. To analyse the amount of isotope fraction potentially driven by temperature variability (thermal diffusion component) we subtracted the fraction of the gravitational fractionation (which is proportional to the LID) derived when fitting $\delta^{15}N$ and $\delta^{40}Ar$ from the respective isotope data. Table 3 shows the outcomes of this analysis. The calculated slope of the unfiltered data ($\delta^{15}N(y)$ and $\delta^{40}Ar/4(x)$) is 1.14. After subtracting the respective gravitational signals from $\delta^{15}N$ and $\delta^{40}Ar$ (grav. corrected) the slope increases towards 1.35, which is still 8% lower than the theoretical estimate. The regression of the gravitational corrected values after low-pass filtering with 100 yr cut-off-period leads to a slope of 1.45, nearly the theoretical value. This result shows that the major long-term trends in $\delta^{15}N$ and $\delta^{40}Ar$ are driven by temperature and that most probably noise has a major impact on multi-decadal data (T < 100 yr). The analysis of the band-pass filtered isotope values shows an equal behaviour. For the multi-centennial band the calculated slope is 1.39 (which is 5% lower than the theoretical value). For the multi-millennial band the slope almost equals the theoretical expectation.





| Periodic-time band | | r | slope |
|---|---|---|---|
| multi-decadal (50-200 yr) | uncorrected | 0.68 | 0.89 ± 0.05 |
| | grav. corrected | 0.68 | 0.89 ± 0.05 |
| multi-centennial (200 yr-1 kyr) | uncorrected | 0.93 | 1.29 ± 0.04 |
| | grav. corrected | 0.91 | 1.39 ± 0.04 |
| multi-millennial (1 kyr-4 kyr) | uncorrected | 0.93 | 1.14 ± 0.03 |
| | grav. corrected | 0.82 | 1.44 ± 0.06 |
| unfiltered data | uncorrected | 0.98 | 1.14 ± 0.02 |
| | grav. corrected | 0.91 | 1.35 ± 0.04 |
| | grav. corrected (low-pass: 100 yr) | 0.97 | 1.45 ± 0.03 |
| theoretical value | | | 1.46 |

**Table 3: Slopes and correlation coefficients between $\delta^{15}N(y)$ and $\delta^{40}Ar/4(x)$ derived using geometric-mean-regression. The theoretical value was calculated as ratio of the thermal-diffusion-sensitivities as $\Omega_N/[\Omega_{Ar}/4]$.**

### 4.3 Model comparison

The comparison of the fitting results using the Schwander et al. (1997) firn-model (black lines, S-Model) and the Goujon et al. (2003) firn-model (blue lines, G-Model) are shown in fig. 12. Figures 12(a-d) contains temperature estimates obtained by fitting the specific isotope targets ((a): $\delta^{15}N$-fitting; (b): $\delta^{40}Ar$-fitting; (c) hybrid solution (see 3.3.); (d) $\delta^{15}N_{excess}$-fitting). In (d) the best-fit solution (red line, bf) is shown additionally due to the large spread of temperature solutions when fitting $\delta^{15}N_{excess}$ using the Schwander et al. (1997) model. The subplots in (a-d) are the differences between the temperature estimates obtained by using the two firn-models displayed as time-series (rose line, thin line shows full resolution case) and histograms. Except for $T(\delta^{15}N_{excess})$, the solutions gained by using the two models show high correlation ($r > 0.9$) in all considered periodic-time bands. Interestingly, fitting of $\delta^{15}N_{excess}$ with the Goujon et al. (2003) model leads to a smaller spread of possible temperature estimates compared to the Schwander et al. (1997) model. The reproducibility between 10 runs (see figs. 5, 12) is 2.7 times better when the Goujon model is used. The reason for that difference was not found yet.

The implementation of the geothermal heat flux in the Goujon model provides a negative constant fraction to $\Delta T_{firn}$ and may lead to this stabilization effect. As in fig.12 the anomalies are shown (relative to 11.5 kyr b2k) it has to be mentioned that the absolute temperatures show an offset of about 2 K between the two models (see histograms). This offset can be explained by the implemented convective zone in the Goujon model. As the convective zone lowers the height of the diffusive firn column a colder temperature (compared to the Schwander



model) is needed, decelerating the densification and leading to the LID needed to fit the gravitational fraction of the gas-isotope data. Therefore the absolute temperature in the Goujon model is 2 K colder compared to the Schwander model. As we see in the time-series and histograms the temperature differences between the estimates using both models are not fully constant. Besides the offset we find mean differences (2σ) of 0.62 K for $T(\delta^{15}N)$ and 0.73 K for $T(\delta^{40}Ar)$ which are partly driven by remaining mismatches of the isotope fits. In the early-Holocene (9.5 kyr to 11.5 kyr) the $T(\delta^{15}N)$ and $T(\delta^{40}Ar)$ reconstructions using the Goujon firn-model show a faster warming, leading to slightly higher temperature anomalies compared to the Schwander model estimates. The rest of the time-series show the reverse behaviour. Here the Schwander model estimates are slightly warmer than the Goujon model estimates. The fact that the trend in the time-series of the differences between the Schwander and Goujon temperature (rose lines) follows the trend in the reconstructed temperature anomalies is pointing to a temperature sensitive fraction of these differences. An explanation for this could be that the densification itself is slightly temperature depended. The absolute temperature in the Goujon model inputs is about 2 K colder as for the Schwander model leading to a different densification pathway. Nevertheless, both models provide highly comparable temperature estimates, which can be seen from the high correlations (tables 7-9) and the agreement of the shapes of many of the shorter term features.

**Figure 12: Temperature solutions for all targets with differences between Schwander and Goujon modelling (histogram in the plots), and mean misfits for all species (histogram right hand side).**





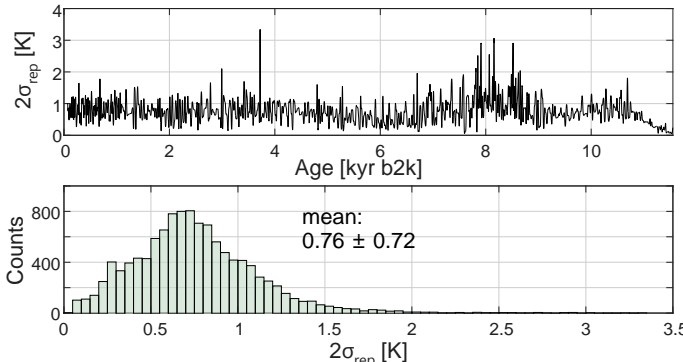

**Figure 13: Reproducibility of temperature solutions between 10 runs using Goujon model and $\delta^{15}N_{excess}$ target. (Reproducibility is more than 2.7 times better as for the Schwander model)**

**4.4 Final uncertainty of reconstructed temperature**

Using all information from the previous sections we are able to calculate an overall limit of the mean uncertainty for reconstructed temperatures using the following equation.

$$(2\sigma_T)^2 = \left(\frac{2\sigma_{miss}}{\Omega_X}\right)^2 + \left(2\sigma_{rep}\right)^2 + \left(\frac{2\sigma_{meas}}{\Omega_X}\right)^2 + \left(2\sigma_{model}\right)^2 \tag{5}$$

$\sigma_T$ is the uncertainty of the reconstructed temperature, $\sigma_{miss}$ is the remaining mean mismatch after fitting the isotope values (sect. 3.2.2), $\sigma_{rep}$ is the fraction of uncertainty due to the reproducibility of the fitting method

(sect. 3.2.1), $\sigma_{meas}$ is the analytical uncertainty of the measured data (tab.1) and $\sigma_{model}$ is the standard deviation of the differences of the temperature estimates between the used models (sect. 4.3.). $\Omega_x$ is the thermal diffusion sensitivity of the respective isotope species. For $T(\delta^{15}N)$ the calculated uncertainty is $2\sigma_T = 0.80\ldots0.88$ K, the range is due to the minimal or maximal analytical uncertainty of the measured data. The uncertainty of the $\delta^{40}Ar$ reconstruction is $2\sigma_T = 0.87\ldots1.81$ K. This final uncertainty is attributed to each single temperature point in time

(the mean data resolution was 17.8 yr), so a smoothing or running mean calculation will decrease the uncertainty due to the averaging over a certain amount of points. For example, a smoothing with 100 yr cut-off will reduce the uncertainty with a factor of 1/sqrt(n) = 1/sqrt(100 yr/17.8 yr) = 0.42 as it equals an averaging over 5.6 points. This is important to keep in mind when discussing filtered versions of these reconstructed temperatures. For $T(\delta^{15}N_{excess})$ we are not able to provide a final uncertainty as we do not understand the reason for the different

behaviour when fitting the data with the two models (stable Goujon- vs. unstable Schwander-solutions). However, as fitting of $\delta^{15}N_{excess}$ leads to highly distinct temperature estimates compared to $T(\delta^{15}N)$ or $T(\delta^{40}Ar)$ and also when compared to other reconstructions (comparison of fig.12(D) and fig.14(red,black)) we do not recommend to use $T(\delta^{15}N_{excess})$ for any climatic interpretation yet.





| | $T(\delta^{15}N)$ S-Model | $T(\delta^{40}Ar)$ S-Model | $T(hyb)$ S-Model | $T(\delta^{15}N_{excess})$ S-Model-bf | $T(\delta^{15}N_{excess})$ S-Model-m | $T(\delta^{15}N)$ G-Model | $T(\delta^{40}Ar)$ G-Model | $T(hyb)$ G-Model | $T(\delta^{15}N_{excess})$ G-Model |
|---|---|---|---|---|---|---|---|---|---|
| $T(\delta^{15}N)$ S-Model | | 0.96 | 0.83 | -0.16 (0.25) | 0.13 (0.31) | 0.99 | 0.95 | 0.79 | 0.31 (0.13) |
| $T(\delta^{40}Ar)$ S-Model | 0.96 | | 0.74 | -0.19 (0.22) | 0.05 (0.42) | 0.95 | 0.98 | 0.70 | 0.26 (0.21) |
| $T(hybrid)$ S-Model | 0.83 | 0.74 | | -0.16 (0.23) | 0.12 (0.34) | 0.81 | 0.71 | 0.98 | 0.57 |
| $T(\delta^{15}N_{excess})$ S-Model-bf | -0.16 (0.25) | -0.19 (0.22) | -0.16 (0.23) | | 0.88 | -0.12 (0.29) | -0.16 (0.25) | -0.21 (0.16) | 0.19 (0.18) |
| $T(\delta^{15}N_{excess})$ S-Model-m | 0.13 (0.31) | 0.05 (0.42) | 0.12 (0.34) | 0.88 | | 0.16 (0.26) | 0.08 (0.39) | 0.06 (0.41) | 0.30 (0.08) |
| $T(\delta^{15}N)$ G-Model | 0.99 | 0.95 | 0.81 | -0.12 (0.29) | 0.16 (0.26) | | 0.95 | 0.78 | 0.36 (0.05) |
| $T(\delta^{40}Ar)$ G-Model | 0.95 | 0.98 | 0.71 | -0.16 (0.25) | 0.08 (0.39) | 0.95 | | 0.67 | 0.27 (0.17) |
| $T(hybrid)$ G-Model | 0.79 | 0.70 | 0.98 | -0.21 (0.16) | 0.06 (0.41) | 0.78 | 0.67 | | 0.53 |
| $T(\delta^{15}N_{excess})$ G-Model | 0.31 (0.13) | 0.26 (0.21) | 0.57 | 0.19 (0.18) | 0.30 (0.08) | 0.36 (0.05) | 0.27 (0.17) | 0.53 | |

**Table 4: correlation coefficient r, statistical significance (p): low-pass cop = 50 yr, if not otherwise stated, p < 0.01.**





| x/y | T($\delta^{15}$N) S-Model | T($\delta^{40}$Ar) S-Model | T(hybid) S-Model | T($\delta^{15}$N$_{excess}$) S-Model-bf | T($\delta^{15}$N$_{excess}$) S-Model-m | T($\delta^{15}$N) G-Model | T($\delta^{40}$Ar) G-Model | T(hybrid) G-Model | T($\delta^{15}$N$_{excess}$) G-Model |
|---|---|---|---|---|---|---|---|---|---|
| T($\delta^{15}$N) S-Model | | 0.67 ¦<br>*0.69 [-4]* | 0.33 ¦ | 0.06 (0.17) ¦<br>*0.13 [-39]* | 0.13 (0.01) ¦<br>*0.18 [-19]* | 0.95<br>*0.97 [-3]* | 0.67 ¦<br>*0.69 [-4]* | 0.33 ¦ | 0.29 ¦<br>*0.31 [-6]* |
| T($\delta^{40}$Ar) S-Model | 0.67 ¦<br>*0.69 [-4]* | | -0.43 ¦<br>*-0.44 [-3]* | 0.25 ¦<br>*-0.41 [-42]* | 0.10 (0.07) ¦<br>*-0.36 [-29]* | 0.69 ¦ | 0.99 ¦ | -0.43 ¦ | 0.12 (0.02) ¦<br>*-0.38 [-18]* |
| T(hybid) S-Model | 0.33 ¦ | -0.43 ¦<br>*-0.44 [-3]* | | -0.23 ¦<br>*0.71 [-39]* | -0.12 (0.05) ¦<br>*0.69 [-25]* | 0.31 ¦ | -0.43 ¦<br>*-0.45 [2]* | 0.96 ¦<br>*0.98 [3]* | 0.63 ¦<br>*0.84 [-11]* |
| T($\delta^{15}$N$_{excess}$) S-Model-bf | 0.06 (0.17) ¦<br>*0.13 [-39]* | 0.25 ¦<br>*-0.41 [-42]* | -0.23 ¦<br>*0.71 [-39]* | | 0.69 ¦<br>*0.95 [11]* | 0.08 (0.10) ¦<br>*0.12 [36]* | 0.27 ¦<br>*-0.41 [42]* | -0.34 ¦<br>*0.74 [43]* | 0.06 ¦<br>*0.79 [24]* |
| T($\delta^{15}$N$_{excess}$) S-Model-m | 0.13 (0.01) ¦<br>*0.18 [-19]* | 0.10 (0.07) ¦<br>*-0.36 [-29]* | -0.12 (0.05) ¦<br>*0.69 [-25]* | 0.69 ¦<br>*0.95 [11]* | | 0.15 ¦<br>*0.18 [13]* | 0.11 (0.05) ¦<br>*-0.37 [29]* | -0.05 (0.25) ¦<br>*0.72 [29]* | 0.59 ¦<br>*0.81 [12]* |
| T($\delta^{15}$N) G-Model | 0.95<br>*0.97 [-3]* | 0.69 ¦ | 0.31 ¦ | 0.08 (0.10) ¦<br>*0.12 [36]* | 0.15 ¦<br>*0.18 [13]* | | 0.70 ¦ | 0.30 ¦ | 0.31 ¦ |
| T($\delta^{40}$Ar) G-Model | 0.67 ¦<br>*0.69 [-4]* | 0.99 ¦ | -0.43 ¦<br>*-0.45 [2]* | 0.27 ¦<br>*-0.41 [42]* | 0.11 (0.05) ¦<br>*-0.37 [29]* | 0.70 ¦ | | -0.45 ¦ | -0.12 (0.03) ¦<br>*-0.40 [-18]* |
| T(hybrid) G-Model | 0.33 ¦ | -0.43 ¦ | 0.96 ¦<br>*0.98 [3]* | -0.34 ¦<br>*0.74 [43]* | -0.05 (0.25) ¦<br>*0.72 [29]* | 0.30 ¦ | -0.45 ¦ | | -0.47 ¦<br>*0.83 [-15]* |
| T($\delta^{15}$N$_{excess}$) G-Model | 0.29 ¦<br>*0.31 [-6]* | 0.12 (0.02) ¦<br>*-0.38 [-18]* | 0.63 ¦<br>*0.84 [-11]* | 0.06 ¦<br>*0.79 [24]* | 0.59 ¦<br>*0.81 [12]* | 0.31 ¦ | -0.12 (0.03) ¦<br>*-0.40 [-18]* | -0.47 ¦<br>*0.83 [-15]* | |

**Table 5: correlation coefficient r, statistical significance (p) ¦ maximum correlation coefficient for cross-correlation *xcf*(x/y) on respective [lag], unit of [lag] is yr. band-pass: 50-200yr (multi-decadal), if not otherwise stated, p < 0.01.**



| | $T(\delta^{15}N)$ S-Model | $T(\delta^{40}Ar)$ S-Model | $T(hybrid)$ S-Model | $T(\delta^{15}N_{excess})$ S-Model-bf | $T(\delta^{15}N_{excess})$ S-Model-m | $T(\delta^{15}N)$ G-Model | $T(\delta^{40}Ar)$ G-Model | $T(hyb)$ G-Model | $T(\delta^{15}N_{excess})$ G-Model |
|---|---|---|---|---|---|---|---|---|---|
| $T(\delta^{15}N)$ S-Model | | 0.87 | 0.59 | 0.54 | 0.56 | 0.96 | 0.87 | 0.51 | 0.59 |
| $T(\delta^{40}Ar)$ S-Model | 0.87 | | 0.18 (0.05) | 0.20 (0.03) | 0.19 (0.05) | 0.82 | 0.94 | 0.17 (0.07) | 0.19 (0.05) |
| $T(hybrid)$ S-Model | 0.59 | 0.18 (0.05) | | 0.73 | 0.85 | 0.63 | 0.24 (0.02) | 0.95 | 0.96 |
| $T(\delta^{15}N_{excess})$ S-Model-bf | 0.54 | 0.20 (0.03) | 0.73 | | 0.97 | 0.58 | 0.27 | 0.54 | 0.87 |
| $T(\delta^{15}N_{excess})$ S-Model-m | 0.56 | 0.19 (0.05) | 0.85 | 0.97 | | 0.60 | 0.26 (0.01) | 0.70 | 0.95 |
| $T(\delta^{15}N)$ G-Model | 0.96 | 0.82 | 0.63 | 0.58 | 0.60 | | 0.88 | 0.57 | 0.63 |
| $T(\delta^{40}Ar)$ G-Model | 0.87 | 0.94 | 0.24 (0.02) | 0.27 | 0.26 (0.01) | 0.88 | | 0.23 (0.02) | 0.25 (0.01) |
| $T(hybrid)$ G-Model | 0.51 | 0.17 (0.07) | 0.95 | 0.54 | 0.70 | 0.57 | 0.23 (0.02) | | 0.84 |
| $T(\delta^{15}N_{excess})$ G-Model | 0.59 | 0.19 (0.05) | 0.96 | 0.87 | 0.95 | 0.63 | 0.25 (0.01) | 0.84 | |

**Table 6 correlation coefficient r, statistical significance (p): band-pass: 200-1000yr (multi-centennial), if not otherwise stated, p < 0.01.**

| | $T(\delta^{15}N)$ S-Model | $T(\delta^{40}Ar)$ S-Model | $T(hybrid)$ S-Model | $T(\delta^{15}N_{excess})$ S-Model-bf | $T(\delta^{15}N_{excess})$ S-Model-m | $T(\delta^{15}N)$ G-Model | $T(\delta^{40}Ar)$ G-Model | $T(hybrid)$ G-Model | $T(\delta^{15}N_{excess})$ G-Model |
|---|---|---|---|---|---|---|---|---|---|
| $T(\delta^{15}N)$ S-Model | | 0.94 | 0.75 | 0.61 | 0.68 | 0.97 | 0.85 | 0.64 | 0.66 |
| $T(\delta^{40}Ar)$ S-Model | 0.94 | | 0.70 | 0.48 | 0.54 | 0.91 | 0.90 | 0.64 | 0.54 |
| $T(hybrid)$ S-Model | 0.75 | 0.70 | | 0.86 | 0.89 | 0.74 | 0.60 | 0.97 | 0.89 |
| $T(\delta^{15}N_{excess})$ S-Model-bf | 0.61 | 0.48 | 0.86 | | 0.96 | 0.67 | 0.46 | 0.78 | 0.98 |
| $T(\delta^{15}N_{excess})$ S-Model-m | 0.68 | 0.54 | 0.89 | 0.96 | | 0.70 | 0.51 | 0.81 | 0.98 |
| $T(\delta^{15}N)$ G-Model | 0.97 | 0.91 | 0.74 | 0.66 | 0.70 | | 0.88 | 0.64 | 0.69 |
| $T(\delta^{40}Ar)$ G-Model | 0.85 | 0.90 | 0.60 | 0.46 | 0.51 | 0.88 | | 0.52 | 0.51 |
| $T(hybrid)$ G-Model | 0.64 | 0.64 | 0.97 | 0.78 | 0.81 | 0.64 | 0.52 | | 0.82 |
| $T(\delta^{15}N_{excess})$ G-Model | 0.66 | 0.54 | 0.89 | 0.98 | 0.98 | 0.69 | 0.51 | 0.82 | |

**Table 7: correlation coefficient r, statistical significance (p): band-pass: 1000-4000yr (multi-millennial, detrended), if not otherwise stated, p < 0.01.**





**4.5 Comparison of T($\delta^{15}$N) with the reconstructions of Kobashi et al. 2017 and Buizert et al. 2018**

Figure 14 shows the comparison of T($\delta^{15}$N) (blue lines, fig. 14(a)) with the reconstructions of Buizert et al. (2018) (red lines) and Kobashi et al. (2017) (black lines). Additionally we show the comparison between the hybrid-temperature (see sect. 3.2., fig.14(A),green lines, fig. 14(b)) with the Kobashi et al. (2017) solution. Due to the boundary effect (see sect. 3.2.3) when using the fitting algorithm (Döring and Leuenberger, 2018) we show for T($\delta^{15}$N) the most reasonable cooling solution (best fit with the borehole temperature profile) for the late-Holocene (last 1 kyr). As stated before, the temperature reconstructions of Kobashi et al. (2017) was conducted using $\delta^{15}$N$_{excess}$ to obtain temperature, which is in our opinion problematic due to the high relative uncertainty of $\delta^{15}$N$_{excess}$ (see sect. 3.1.) and the systematic offset to too high $\delta^{40}$Ar (see sect. 3.2. and sect. 4.1.). Buizert et al. (2018) use the calibration of water-stable-isotope $\delta^{18}$O to fit the long-term trend of $\delta^{15}$N for the early to mid-Holocene (until 4 kyr b2k). From 4 kyr to today they use the temperature from Kobashi et al. (2017) superimposed with a lager cooling trend (see fig.15). First, we compare the major long-term trends using ordinary regression. For the strong warming trend in the early-Holocene (9-11.2 kyr b2k) the Buizert et al. reconstructions shows the largest warming trend with 3.07 K/kyr, whereas the warming trend of the Kobashi et al. reconstruction is about 20% smaller with a value of 2.58 K/kyr. A reason for this could be the flattening of T($\delta^{15}$N$_{excess}$) during this time as discussed in sect. 4.1.(ii). Our $\delta^{15}$N-based reconstruction delivers an intermediate value of 2.75 K/kyr which is about 10% smaller as for the Buizert et al. reconstruction. Next, we compare the mid-Holocene (4.0-8.0 kyr b2k) cooling trends. Here our $\delta^{15}$N-based reconstruction shows the largest cooling trend with a value of -0.33 K/kyr. The Buizert et al. estimate shows about 10% less cooling (0.29 K/kyr) and the Kobashi et al. reconstruction 39% less cooling (-0.24 K/kyr). For the mid- to late-Holocene (1.0-4.0 kyr b2k) the Kobashi et al. reconstructions shows nearly no trend (slightly warming of 0.07 K/kyr) whereas the Buizert et al. (-0.26 K/kyr) and T($\delta^{15}$N) estimates (-0.17 K/kyr) are pointing to a further long-term cooling during this time. Next, we compare the variance of the temperatures (tab. 8) for two time-windows (0.5-4.0 kyr b2k and 4.0-11.2 kyr b2k) and two periodic-time bands (bands: 100 yr to 500 yr and 500 yr to 4 kyr). As the temporal resolution of the Buizert at al. estimate is 20 yr, we resampled our data and the Kobashi et al. data to the same grid before band-pass filtering. Also, we cut out the 8.2k-event, as it dominates the variance of the temperature data. In the early- to mid-Holocene (4.0-11.2 kyr b2k, w/o 8,2k-event), the standard deviation (2σ) of T($\delta^{15}$N) in the 100-500 yr periodic-time band is 0.47 K which is nearly equal to the Buizert at al. reconstruction with a value of 0.49 K. The Kobashi et al. reconstructions has a more than double as large variance in that section with 2σ = 1.17 K. For the mid- to late-Holocene (0.5-4.0 kyr b2k), the standard deviation of T($\delta^{15}$N) is about 20% smaller (2σ = 0.37 K) than for the early- to mid-Holocene (4.0-11.2 kyr). Also the variance of the Kobashi et al. reconstruction is slightly smaller (17%, 2σ = 0.97 K) during this time. The Buizert et al. reconstruction nearly equals the Kobashi et al. estimate here, with 2σ = 1.00 K. This is not unexpected as Buizert et al. uses the Kobashi et al. data with slightly modifications. But it is not quite reasonable that a doubling of the variance between the two parts of the Holocene is realistic. For longer periodicities (band: 0.5-4.0 kyr) we also see a reduction of the variance for the late-Holocene compared to the early-Holocene in all three reconstructions. During the early- to mid-Holocene the variance of T($\delta^{15}$N) and the Buizert et al. reconstruction are in good agreement (2σ$_{T(\delta 15N)}$ = 0.58 K, 2σ$_{Buizert}$ = 0.65 K) whereas the Kobashi et al. estimate is pointing to higher variability (2σ$_{Kobashi}$ = 1.26 K), an equal behaviour as it was found for the faster periodicities. For the mid- to late-Holocene T($\delta^{15}$N) shows the smallest variability (2σ = 0.35 K) compared to the Buizert at al. (2σ = 0.58 K) and Kobashi et al. (2σ = 0.81 K) reconstructions. We conducted the same analysis for the accumulation-rate





input data (the 200 km scenario) and for the gas-isotope data (tab.9). Especially for the 0.5-4.0 kyr periodic-time band, we find an equal reduction of the data variance between the early- to mid- and the mid- to late-Holocene. For $\delta^{15}N$ the standard deviation ($2\sigma$) in the early- to mid-Holocene is about 52% higher as in the mid- to late-Holocene, 34% for $\delta^{40}Ar$ and 41% for $\delta^{15}N_{excess}$, respectively. Also the accumulation-rate data show this

deviation of the variance between the two time sections in this periodic-time band with a reduction of 46%. For the faster periodicities (band-pass 100-500 yr), the gas-isotope data also shows that behaviour but with less disagreement between the two time sections. In contrast to that the accumulation-rate data show a slightly higher variance in the mid- to late-Holocene if compared to the early to mid-Holocene. Based on those findings we conclude that the difference in the variability of our temperature estimates between the early- to mid- and the

mid- to late-Holocene is a direct result of that behaviour of the gas-isotope target data and the accumulation-rate input. Interesting is the difference between the Buizert et al. and Kobashi et al. estimates for the time section 0.5-4.0 kyr b2k, pointing to the necessity of modifications on the Kobashi et al. estimate when used in the study of Buizert et al. (2018). A reason for that could be the use of a different firn-model or possible memory effects occurring due to the differences in the temperature estimates between the Buizert et al. and Kobashi et al.

estimates in the early to mid-Holocene. Figure 14(b) shows the comparison between the Kobashi et al. estimate and our hybrid temperature. The hybrid temperature was calculated (see sect. 3.2.) from the long-term temperature trends of $T(\delta^{15}N)$ (low-pass, cop = 500 yr) superimposed with the short-term temperature trends (high-pass, cop = 500 yr) of $T(\delta^{15}N_{excess})$. It is obvious that both estimates show a good agreement, especially for the faster features (tab. 10). In four time sections we find larger offsets. These sections are: 9-10.5 kyr, shortly

after the 8.2k-event (6.6-8.1 kyr), 5.3-6.1 kyr and 0.07-1.8 kyr. All sections start with a fast warming or cooling trend in the Kobashi et al. estimate with duration of about 100-200 yr. It is highly probable that these shifts are introduced by the "window correction method" used by Kobashi et al. (2017), where the fitting target ($\delta^{15}N_{excess}$, $\Delta T_{firn}$) are slightly shifted with certain offsets in six time-windows of 1500 yr length to improve the temperature estimate compared to the pure $\delta^{15}N_{excess}$-based reconstruction, which leads to unrealistic temperature estimates if

not corrected as shown in fig.11(A) or fig.12(D). The goal of that correction was to change $T(\delta^{15}N_{excess})$ such that the general trend in $\delta^{15}N$ is matched. In our view, this correction method is highly critical as the choice of the window-length, the window positions and the found offsets are more or less arbitrary, but on the other side crucial for the reconstruction. Additionally, we correlated the Buizert et al. (2018) and Kobashi et al. (2017) temperature estimates with each other and with our data after low-pass filtering (cop = 50 yr) and in all three

investigated periodic-time bands (multi-decadal, multi-centennial, multi-millennial, tab. 10). If not otherwise stated the statistical significance is always p < 0.01. For the Buizert et al. reconstruction the correlations were calculated only for the early- to mid-Holocene values (4.0-11.5 kyr b2k), whereas the correlations between our and the Kobashi et al. estimates were calculated for 0.5-11.5 kyr b2k. For the low-pass filtered values (general trend) the Buizert et al. reconstruction shows the highest correlations with $T(\delta^{15}N)$ and $T(\delta^{40}Ar)$ with r > 0.9 and

a correlation of r = 0.82 with the Kobashi et al. estimate due to the high agreement in the general trend. The comparison between the Kobashi et al. temperature and our estimates has the highest correlation for the hybrid temperatures (r = 0.87, r = 0.83), and $T(\delta^{15}N)$ (r = 0.81) in that case. Besides the general trend the correlations become reduced for faster oscillations. In all periodic-time bands the Buizert et al. reconstruction has the highest correlation with $T(\delta^{15}N)$ with $r_m = 0.67$, $r_c = 0.61$ and $r_d = 0.30$ for multi-millennial, multi-centennial and multi-

decadal signals, respectively. The agreement between the Buizert et al. and the Kobashi et al. reconstructions are $r_m = 0.52$ (p = 0.03), $r_c = 0.48$ and $r_d = 0.05$ (p = 0.25). That finding implies that the agreement between the





Buizert et al. $\delta^{18}O_{ice}$-based and our $\delta^{15}N$-based reconstruction is generally higher as for the $\delta^{18}O_{ice}$-based reconstruction compared to the $\delta^{15}N_{excess}$-based one of Kobashi et al.. On the other hand the correlations are relatively weak (but still significant) especially for the multi-decadal band.

| time section [kyr b2k] | Band-pass: 100 yr-500 yr | | | Band-pass: 0.5 kyr-4.0 kyr | | |
|---|---|---|---|---|---|---|
| | $2\sigma$ [K] $T(\delta^{15}N)$ | $2\sigma$ [K] Buizert 2018 | $2\sigma$ [K] Kobashi 2017 | $2\sigma$ [K] $T(\delta^{15}N)$ | $2\sigma$ [K] Buizert 2018 | $2\sigma$ [K] Kobashi 2017 |
| 0.5-11.2 w/o 8.2k-event | 0.44 | 0.72 | 1.10 | 0.51 | 0.63 | 1.12 |
| 0.5-4.0 | 0.37 | 1.00 | 0.97 | 0.35 | 0.58 | 0.81 |
| 4.0-11.2 w/o 8.2k-event | 0.47 | 0.49 | 1.17 | 0.58 | 0.65 | 1.26 |

**Table 8: Standard deviations ($2\sigma$) of temperature estimates of $T(\delta^{15}N)$ (this study), Buizert et al. (2018) and Kobashi et al. (2017) when band-pass filtered for two periodic-time bands and calculated without the 8.2k-event.**

| time section [kyr b2k] | Band-pass: 100 yr-500 yr | | | | Band-pass: 0.5 kyr-4.0 kyr | | | |
|---|---|---|---|---|---|---|---|---|
| | $2\sigma$ [permeg] $\delta^{15}N$ | $2\sigma$ [permeg] $\delta^{40}Ar$ | $2\sigma$ [permeg] $\delta^{15}N_{excess}$ | $2\sigma$ [mm/yr] acc | $2\sigma$ [permeg] $\delta^{15}N$ | $2\sigma$ [permeg] $\delta^{40}Ar$ | $2\sigma$ [permeg] $\delta^{15}N_{excess}$ | $2\sigma$ [mm/yr] acc |
| 0.5-11.2 w/o 8.2k-event | 7.2 | 6.9 | 5.2 | 11.7 | 6.6 | 5.5 | 3.0 | 8.2 |
| 0.5-4.0 | 6.6 | 5.9 | 3.9 | 13.0 | 3.7 | 4.0 | 2.0 | 5.1 |
| 4.0-11.2 w/o 8.2k-event | 7.5 | 7.3 | 5.7 | 11.0 | 7.7 | 6.1 | 3.4 | 9.4 |

**Table 9: Standard deviations ($2\sigma$) of gas-isotope measured data and accumulation-rates when band-pass filtered for two periodic-time bands and calculated without the 8.2k-event.**



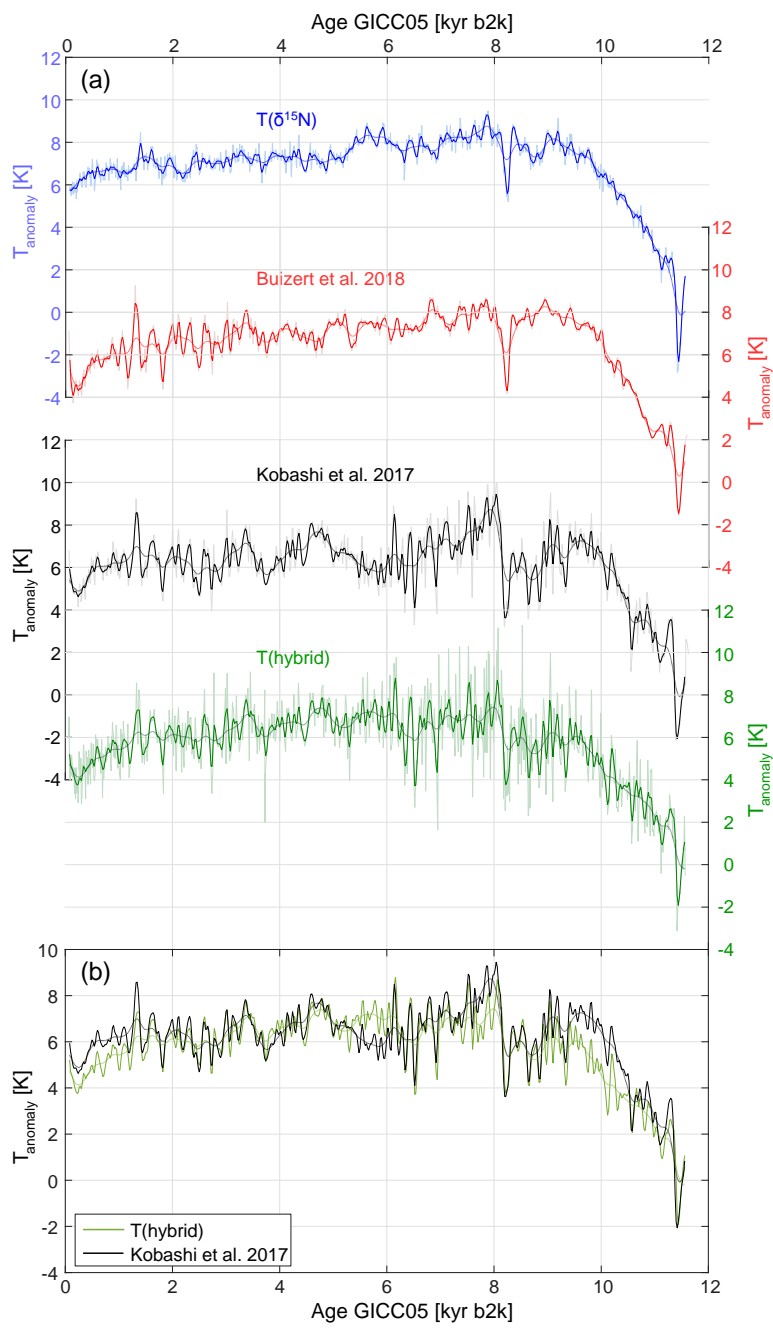

**Figure 14: (A) Comparison of T(δ[15]N) (blue lines) and T(hybrid) (green lines) modelled using the Schwander et al.
model with the temperature reconstructions (anomalies rel. to 11.500 kyr b2k) for the GISP2 site from Buizert et al.
2018 (red lines) and Kobashi et al. 2017 (black lines). Thin lines show unfiltered data, thick lines show low-pass
filtered data using cop = 100 yr, dotted lines show low-pass filtered data using cop = 500 yr. (B) Comparison between
the reconstruction from Kobashi et al. 2017 (black lines) and T(hybrid) (lines) in the same plot.**





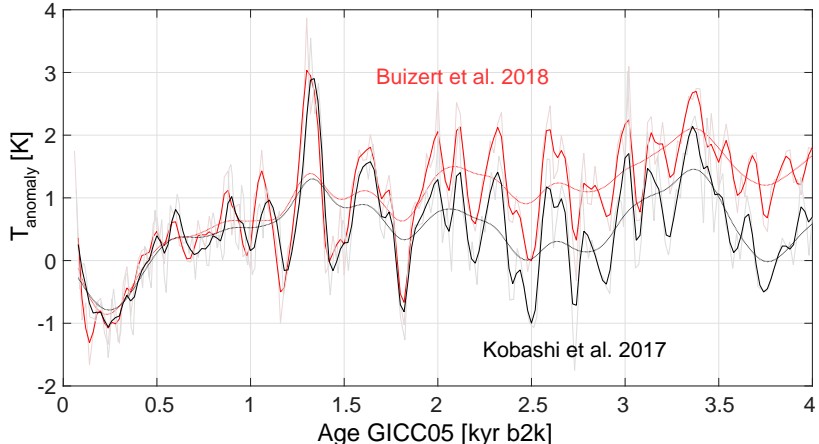

**Figure 15: Comparison of the reconstructed temperatures anomalies (rel. to last 1 kyr) of Buizert et al. (2018) (red lines) and Kobashi et al. (2017) (black lines) for the last 4 kyr. Thick lines show data smooth with 100 yr cut-off, dotted line with 500 yr cut-off, thin lines are 20 yr resolution.**





| | Kobashi et al. 2017 ($T_K$)* | | | | Buizert et al. 2018 ($T_B$)** | | | |
|---|---|---|---|---|---|---|---|---|
| | low-pass cop [yr] 50 | band-pass [yr] 50-200 (multi-decadal) | band-pass [yr] 200-1000 (multi-centennial) | band-pass [yr] 1000-4000 (multi-millenial) | low-pass cop [yr] 50 | band-pass [yr] 50-200 (multi-decadal) | band-pass [yr] 200-1000 (multi-centennial) | band-pass [yr] 1000-4000 (multi-millenial) |
| $T(\delta^{15}N)$ S-Model | 0.81 | 0.21 ¦ <br> *0.26* <br> [12] | 0.61 | 0.50 | 0.95 | 0.22 ¦ <br> *0.30* <br> [17] | 0.61 | 0.67 |
| $T(\delta^{40}Ar)$ S-Model | 0.72 | -0.25 ¦ <br> (0.01) <br> *-0.37* <br> [12] | 0.25 | 0.46 | 0.92 | 0.24 ¦ <br> *0.26* <br> [10] | 0.54 | 0.63 |
| $T(hybrid)$ S-Model | 0.87 | 0.67 ¦ <br> *0.81* <br> [9] | 0.90 | 0.78 | 0.79 | -0.01 <br> (0.44) ¦ | 0.39 | 0.50 <br> (0.01) |
| $T(\delta^{15}N_{excess})$ S-Model-bf | 0.02 <br> (0.47) | -0.06 <br> (0.20) ¦ <br> *0.67* <br> [-30] | 0.79 | 0.78 | -0.22 <br> (0.31) | -0.05 <br> (0.25) ¦ | 0.38 | 0.35 <br> (0.07) |
| $T(\delta^{15}N_{excess})$ S-Model-m | 0.30 <br> (0.08) | 0.42 ¦ <br> *0.68* <br> [-16] | 0.87 | 0.76 | 0.15 <br> (0.38) | 0.06 <br> (0.19) ¦ | 0.38 | 0.48 |
| $T(\delta^{15}N)$ G-Model | 0.82 | 0.24 ¦ <br> *0.27* <br> [9] | 0.66 | 0.51 | 0.93 | 0.26 ¦ <br> *0.29* <br> [11] | 0.63 | 0.57 |
| $T(\delta^{40}Ar)$ G-Model | 0.70 | -0.25 ¦ <br> *-0.37* <br> [12] | 0.30 | 0.36 <br> (0.06) | 0.90 | 0.24 ¦ <br> *0.26* <br> [10] | 0.61 | 0.45 <br> (0.04) |
| $T(hybrid)$ G-Model | 0.83 | 0.55 ¦ <br> *0.82* <br> [13] | 0.80 | 0.79 | 0.75 | -0.02 <br> (0.42) ¦ | 0.35 | 0.42 <br> (0.06) |
| $T(\delta^{15}N_{excess})$ G-Model | 0.59 | 0.73 ¦ <br> *0.74* <br> [-3] | 0.92 | 0.78 | 0.31 <br> (0.02) | 0.06 <br> (0.18) ¦ | 0.40 | 0.45 <br> (0.02) |
| $T_K$ | | | | | 0.82** | 0.05** <br> (0.25) ¦ | 0.48** | 0.52** <br> (0.03) |
| $T_B$ | 0.82** | 0.05** <br> (0.25) ¦ | 0.48** | 0.52** <br> (0.03) | | | | |

**Table 10: Correlation coefficient r, statistical significance (p) ¦ maximum correlation coefficient for cross-correlation *xcf*(x/y) on respective [lag], if not otherwise stated: p < 0.01. *The correlations between the Kobashi et al. (2017) and our data were calculated in the time-window 0.5-11.5 kyr b2k. **The correlations between the Buizert et al. (2018) and our data as well as between the Kobashi et al. (2017) and the Buizert et al. (2018) data were calculated in time-window 4.0-11.5 kyr b2k.**





## 5 Conclusion

In this study we applied the Döring and Leuenberger (2018) gas-isotope fitting-algorithm to Holocene $\delta^{15}N$, $\delta^{40}Ar$ and $\delta^{15}N_{excess}$ data measured on the GISP2 ice core (Kobashi et al., 2008b) using two state of the art firn-densification and heat-diffusion models (Goujon et al., 2003; Schwander et al., 1997). The results of this study

are summarized as follows:

*Signal-to-noise study:*

As starting point, a signal-to-noise (SNR) study was conducted investigating the suitability of the three gas-isotope tracers ($\delta^{15}N$, $\delta^{40}Ar$ and $\delta^{15}N_{excess}$) for temperature reconstructions in context of the in Kobashi et al. (2008b) stated measurement uncertainty. It was shown that $\delta^{15}N$ is most favoured to reconstruct Holocene

temperature due to its higher SNR compared to $\delta^{40}Ar$ and $\delta^{15}N_{excess}$ especially for multi-decadal to multi-centennial signals.

*Gas-isotope fitting results and uncertainties:*

To evaluate the performance of the Döring and Leuenberger (2018) gas-isotope fitting-algorithm on measured Holocene data and to constrain the uncertainty of the reconstructed temperatures the reproducibility of the fitting

algorithm was tested using the Schwander et al. (1997) firn-model. The results are showing excellent performance for $\delta^{15}N$ and $\delta^{40}Ar$-fits. The reproducibility between ten runs of the fitting algorithm was quantified to be (mean $\pm$ 2$\sigma$) 1.68 $\pm$ 1.14 permeg for $\delta^{15}N$, 2.58 $\pm$ 1.77 permeg for $\delta^{40}Ar/4$, and 1.28 $\pm$ 2.14 permeg for $\delta^{15}N_{excess}$ for each single point in time. The translation in temperature also shows good reproducibility with (0.17 $\pm$ 0.12 K) for T($\delta^{15}N$) and (0.26 $\pm$ 0.18 K) for T($\delta^{40}Ar$). In contrast the spread for $\delta^{15}N_{excess}$ temperature was

about ten times higher (2.04 $\pm$ 1.90 K) due to the loss of information about the firn-column-height when calculating $\delta^{15}N_{excess}$ which leads to a variety of different possible temperature solutions fitting $\delta^{15}N_{excess}$ with the same precision. During the reproducibility study we noticed a boundary problem emerging at the end of the data set when the data were fitted using both models. To overcome this issue an additional constrain is need, namely the fitting of the measured borehole temperature profile using the Goujon et al. (2003) firn-model. Next, the

contribution of remaining misfits to the uncertainty budget of the reconstructed temperatures was quantified. We reach final misfits (2$\sigma$) of 3.7 permeg for $\delta^{15}N$, 2.8 permeg for $\delta^{40}Ar/4$, and 2.1 permeg for $\delta^{15}N_{excess}$ using the Schwander et al. (1997) model, and 6.0 permeg, 7.0 permeg and 2.0 permeg for $\delta^{15}N$, $\delta^{40}Ar/4$ and $\delta^{15}N_{excess}$ respectively using the Goujon et al. (2003) model. Additionally the influence of three different accumulation-rate estimates on the temperature reconstructions was investigated. The accumulation-rate uncertainty leads to

asynchronous multi-decadal temperature signals due to deviations in the modelled $\Delta$age regimes with maximum differences of 30 yr in the early-Holocene but keeping the amplitudes of the signals unchanged. From the mid-Holocene until today the modelled $\Delta$age difference becomes less than 5 yr. For longer term temperature trends, the higher accumulation-rate scenario leads to a 0.3 K larger cooling compared to the lower and intermediate accumulation-rate scenarios.

*Comparison of model results:*

Next, we compared the temperature estimates calculated by fitting $\delta^{15}N$, $\delta^{40}Ar$ and $\delta^{15}N_{excess}$ with each other. We found significant and high correlation between T($\delta^{15}N$) and T($\delta^{40}Ar$) for multi-centennial and multi-millennial signals, but T($\delta^{40}Ar$) points to higher amplitudes for some of these temperature anomalies. For multi-decadal signals the correlation between T($\delta^{15}N$) and T($\delta^{40}Ar$) is weak but still significant and equals the correlation

between the isotope data ($\delta^{15}N$ and $\delta^{40}Ar$). The comparison of the temperature gradient over the diffusive firn column $\Delta T_{firn}$ modelled from T($\delta^{15}N$) and T($\delta^{40}Ar$) with the $\Delta T_{firn,meas}$ – calculated from the measured $\delta^{15}N_{excess}$ –



documents a good agreement of the general trend in the late- to mid-Holocene (1.3 kyr-6.4 kyr b2k). However, in the early to mid-Holocene (6.4 kyr-11.5 kyr) and the late-Holocene (0.07 kyr-1.3 kyr b2k) $\Delta T_{firn}$ modelled from $\delta^{15}N$ and $\delta^{40}Ar$ significantly exceeds $\Delta T_{firn,meas}$ which might be an indication for systematic too high $\delta^{40}Ar$ in this section potentially caused by post-coring gas-loss. The temperature calculated by fitting $\delta^{15}N_{excess}$ differs

significantly from the coherent $T(\delta^{15}N)$ and $T(\delta^{40}Ar)$, especially for the early and late Holocene. The analysis of these differences also suggests that the $\delta^{40}Ar$ values are too high in these sections. The correlations between $T(\delta^{15}N_{excess})$ with $T(\delta^{15}N)$ or $T(\delta^{40}Ar)$ are weak for all three analysed periodic-time bands (multi-decadal, multi-centennial, multi-millennial). For multi-decadal signals we find a weak negative correlation between $T(\delta^{15}N_{excess})$ and $T(\delta^{40}Ar)$ and no correlation between $T(\delta^{15}N_{excess})$ and $T(\delta^{15}N)$ which implies that the multi-decadal

oscillations in $T(\delta^{15}N_{excess})$ are mainly driven by $\delta^{40}Ar$ with less influence from $\delta^{15}N$ due to a higher noise contribution on $\delta^{40}Ar/4$ compared to $\delta^{15}N$. In addition, we calculated the slope between $\delta^{15}N$ and $\delta^{40}Ar$ using geometric-mean-regression for all three periodic-time bands after subtracting the gravitational components of $\delta^{15}N$ and $\delta^{40}Ar$. Especially for multi-decadal signals the slope significantly underestimates the theoretical value calculated using the empirical derived thermal sensitivities of $\delta^{15}N$ and $\delta^{40}Ar$ by 53%. These results are pointing

to too high $\delta^{40}Ar$ that may be influenced by noise or some kind of fractionation which is not captured by the firn-models (e.g. gas loss). For the multi-centennial and the multi-millennial bands the slope almost equals the theoretical expectation.

*Comparison of model results of two different firn-models:*

Next, we compared reconstructed temperatures obtain by using the Schwander et al. (1997) firn-model with the

solutions of the Goujon et al. (2003) model. For $T(\delta^{15}N)$ and $T(\delta^{40}Ar)$ the temperature estimates show high correlation (r > 0.9) in all considered periodic-time bands (multi-decadal, multi-centennial, multi-millenial) and in amplitudes and shapes of many of the shorter term features. In the early-Holocene (9.5 kyr to 11.5 kyr) the $T(\delta^{15}N)$ and $T(\delta^{40}Ar)$ reconstructions using the Goujon firn-model show a faster warming, leading to slightly warmer temperature anomalies compared to the Schwander model estimates. The rest of the time-series show the

reverse behaviour. Here the Schwander model estimates are slightly warmer than the Goujon model estimates. The result that the difference in the temperature anomalies between both models follows the general trend of the temperature anomalies themselves is pointing to a temperature dependence which is attributed to the temperature dependence of the densification due to the difference in absolute temperatures of about 2 K. The variance of the differences (2σ) between the temperature anomalies obtained by using both models were quantified to be 0.62 K

for $T(\delta^{15}N)$ and 0.73 K for $T(\delta^{40}Ar)$.

*Uncertainty estimation:*

Using all results presented in this study we estimated the mean uncertainty for $T(\delta^{15}N)$ and $T(\delta^{40}Ar)$. The final uncertainty budget of the reconstructed temperature anomalies is dependent on four terms: (i) the mismatch between the measured and modelled isotope data; (ii) the reproducibility of the isotope fits; (iii) the measurement

uncertainty of the isotope data; and (iv) the difference between the temperature estimates using different firn-models (here between two firn-models). Adding up these terms leads to a final uncertainty of $2\sigma_T = 0.80\ldots0.88$ K for $T(\delta^{15}N)$, and $2\sigma_T = 0.87\ldots1.81$ K for $T(\delta^{40}Ar)$.

*Comparison to other published temperature reconstructions:*

Finally, we compared our temperature estimates to temperature reconstructions for the GISP2 site of Buizert et

al. (2018) and Kobashi et al. (2017). First, the variance of the temperature anomalies was analysed in two periodic-time bands (100-500 yr and 0.5-4 kyr). We found a high agreement of the variance between $T(\delta^{15}N)$ and



the Buizert et al. (2018) estimate in both considered bands for the time 4.0-11.5 kyr b2k. In contrast the variance of the Kobashi et al. (2017) temperature anomalies is nearly twice as high as our $T(\delta^{15}N)$ or the Buizert et al. (2018) estimate. Interestingly, all three reconstructions are pointing to a decrease of the variance in the mid- to late-Holocene (0.5-4.0 kyr b2k) compared to the early- to mid-Holocene (4.0-11.5 kyr b2k) for multi-centennial to multi-millennial signals. For our reconstruction this result is attributed to an equal behaviour of the gas-isotope and the accumulation-rate data. Finally, we compared the correlations between the three reconstructions for three periodic-time bands (multi-decadal, multi-centennial, multi-millennial). We find generally higher agreement in all bands between our $\delta^{15}N$-based reconstruction and the $\delta^{18}O$- and $\delta^{15}N$-based reconstruction of Buizert et al. (2018) as between our $T(\delta^{15}N)$ estimate and the Kobashi et al. (2017) reconstruction.

**Competing interests**

The authors declare that they have no competing financial interests.

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
