# Peer review of "Comparison of Holocene temperature reconstructions based on GISP2 multiple-gas-isotope measurements"

_Climate of the Past, 2019_

## Referee Comment (RC1) · Anonymous Referee #1 · 18 Dec 2019

Scientific significance: Poor

Does the manuscript represent a substantial contribution to scientific progress within the scope of Climate of the Past (substantial new concepts, ideas, methods, or data)?

This manuscript uses already published data (d15N and d40Ar at GISP2 over the holocene), but a different algorithm (also already published by the authors) to reconstruct the surface air temperature. Although this algorithm presents some advantages, it does not lead to a better fit to the data, and the temperature reconstruction is not better than the 2 already published, and does not provide any new insight on climate.

Scientific quality: poor

[Figure]

Are the scientific approach and applied methods valid? Are the results discussed in an appropriate and balanced way (consideration of related work, including appropriate references)?

The consideration of related work is appropriate. The method is valid in theory, but the implementation has some serious flaws.

The problem here is essentially the resolution of a sytem of 2 equations with 2 unknowns. We have 2 datasets : d15N and d40Ar, plus a third one which is the borehole temperature profile, and 2 input parameters: temperature and accumulation rate. The data (d15N and d40Ar) can be computed to derive 2 variables : the lock-in depth (LID) and the firn temperature gradient DeltaT. A firn densification/heat diffusion model can be use to relate the climate variables (T and acc) to the LID and DeltaT. The authors use 2 good densification models, but strangely, they decide not to use both datasets together in the inverse problem. So they are left with an under-determined system. They are proud to fit the data perfectly, but it is always possible with an under-determined system, and they do not find a coherent answer when they fit different targets (d15N, d40Ar, d15Nexcess). This means that the problem is not well posed. If you have data, you should use them. As a result, the reconstructed temperature makes no sense, especially for d15N excess, but also for the others, as is visible in figure 8 g, h, and i, where the output of the inverse model produces solutions that do not match the borehole temperature data, or known LID (which is unfortunately not shown).

Presentation quality: fair

Are the scientific results and conclusions presented in a clear, concise, and well-structured way (number and quality of figures/tables, appropriate use of English language)?

The text is long, sentences are sometimes very complicated with improper english. I recomment a full rewrite of the paper when the scientific objectives are better defined. General comment on the form of the figures: it would be much better to have the time

axis going always the same way. Figure 9 is the worst, with 2 different directions in the subplots of the same figure.

As a result, I suggest that the paper be rejected, and rewritten for future submission.

The work is not entirely uninteresting, but as it stands, it looks like a student project, and does not add to the litterature.

If I understand the author's motivation correctly, they wanted to use their inverse method to better quantify the uncertainties in temperature reconstructions from inert gases. This is still a valid goal in my mind, but it needs to be tackled differently from what is being done here.

First, the authors should adopt the thinking framework of considering gravitational and thermal fractionation separately, and evaluating both of them. It would avoid the tragedy of Figure 7 or 8h, where inapropriate firn thicknesses are derived.

Regarding gravitational fractionation, here are some questions that are worth investigating:

1. How well is it known from the data? What is the impact of the uncertainty in gas loss on the LID estimation? What is a reasonnable uncertainty range when a convective zone is considered? The output of such a section would give you a LID with uncertainty bars that represent not only the analytical uncertainty, but also other sources, that are more tricky to quantify.

2. Can the densification models accurately reproduce LID variability? This is a big question. Lundin et al 2017 have shown that there are substantial offsets in the mean steadystate LID already, this is why you need a T offset between the 2 models. But what about the variability? If you use a reasonnable range of temperature and accumulation rate change, can you reproduce the high frequency variability in the LID? I bet you will not be able to get a perfect fit, and you'll find that the LID changes more slowly in the model that what the data suggest.

What is shown on Figure 8 g and H is the compensation between the LID and DeltaT. So all the uncertainty in our knowledge of densification is automatically compensated by a temperature adjustment so that thermal fractionation compensates the misfit in gravitational fractionation when you fit d15N only. This is an essential point, that is causing you a lot of trouble, but you have not discussed it, or even shown a model data comparison with LID and DeltaT data. You approach supposes that the firn densification model is perfect, and that is a difficult assumption to make.

Temperature influences both gravitational fractionation (through its influence on firn thickness) and thermal fractionation. Kobashi et al., 2008 have already shown that d15Nexcess alone cannot be faithfully used to reconstruct temperature change, because of the problem with drift. It is not new. Your hybrid solution is a good idea, but why not also use the borehole temperature record, or compare with d18O, which is also sensitive to temperature? Here several options are possible. But what you did does not include enough constraints to give a meaningful result. In theory, your inverse method can be used to fit multiple datasets at the same time. I suggest that you would do that.

Finally, there are some interesting differences between the Kobashi and the Buizert temeprature reconstructions, particularly around 8.2ka, where Kobashi has a big overshoot after 8ka, but Buizert does not. It would be interesting to understand where these differences come from. It would also be interesting to see whether using their temperature histories with the two densification models, you find a good fit to the data. By comparing the different reconstructions, and the d18O record, you could have a more in-depth discussion about climate history at GISP2. It would give more meat to the paper, and justify its presence in CP.

Tu sum up, as it stands, this paper does not reach substantial conclusions that justify its publication, but with a bit of work, the authors could submit a new manuscript on this general topic.

---

## Referee Comment (RC2) · Anonymous Referee #2 · 19 Dec 2019

Review for "Comparison of Holocene temperature reconstructions based on GISP2 multiple-gas-isotope measurements", Döring and Leuenberger, 2019, Climate of the Past Discussions

Summary:

Döring and Leuenberger present the first application of a new gas-based temperature inversion procedure to previously published data from GISP2 during the Holocene. This is an interesting and timely contribution to the problem of constraining site temperature from histories of inert gas isotopes that has been gaining traction as an alternative to the classic water isotope paleothermoter. The type of technique they describe is particularly useful as it promises a more subjective and quantitative treatment of uncertainty imparted by the inversion.

Overall, I was overwhelmed by the volume of information provided in the manuscript. I would challenge the authors to consider a significant reduction in length in a new submission. Considering that a separate, lengthy methods paper has already been published, I was slightly surprised to see the volume of additional information required to establish the robustness of their method.

The key questions I had after reading the paper are as follows. What information is crucial and to what accuracy must it be known (e.g. accumulation, bore-hole thermometry, water isotopes temperature reconstructions for initialization, etc.)? What are the knowns that go into the inversions to produce estimates of the unknowns? How over or under determined is the system?

Additionally, there's a thread that runs throughout the paper about the hazards of using 15Nexcess. Because of the length and wide range of topics, the arguments end up being very dispersed. I would have appreciated a more concise structure to this thread as it took me a number of reads to find all the salient points.

Because of the length I have decided not to make line-by-line suggestions. Rather, I've broken the paper down into sections and highlighted what my main "take-home messages" were as a reader and any key questions I had.

**Abstract**
I count five major points in the abstract.
1) Comparison of 15N, 40Ar 15Nex
2) Gas loss as an artefact in 40Ar
3) Comparison of two different densification models
4) Treatment of analytical uncertainty
5) Comparison of Buizert et al., Kobashi et al, with this study

I personally think the paper could be reduced to about five figures along these lines and a few tables; whereas it currently stands at 14 figures and 10 tables.

The use of ellipses (...) in the uncertainty on the temperature is unclear to me until you read deep in into the paper (section 4?)

**1 Introduction**

Please define 15N, 40Ar, and crucially 15Nex. Without stating 15Nex is a function of both 15N and 40Ar it is not clear to the reader that you are in some sense doing a joint inversion when you use 15Nex. Additionally, this would be great time to discuss want information you "lose" when using 15Nex (e.g. LID), which is currently buried deep in the section on signal-to-noise.

Please remove references to a general body of work as simply "Kobashi et al.," or "his work".

**2.1 Firn-models and inversion algorithm**

This is fine but it tends to re-hash the published methods paper. Also, given that the Goujan and Schwander models actually show small difference when fitting 15N and 40Ar, I'm not sure so much detail is needed about basic things like the different zones of densification and depth resolution. It would suffice to say you simply used the two classic firn models.

The point about no geothermal heat flux in the Schwander is an interesting aside – particularly if you were examining a shallow core (<1000 m) - but my take home was that it just imparted a constant offset in the absolute temperature. Given that we are more interested in changes in temperature relative to the modern (when we can actually measure it), I was a bit lost as to why this was so important.

**2.2 Timescale, and necessary data**

Why do the text and nearly all the figures refer to chronology as the "ice age" of GICC05A? Shouldn't it be the gas-age chronology for the gas-based data?

Accumulation-rate input data: the difference in these scenarios needs to be defined. Later on there are many references to the 50km, 100km, etc scenarios but no information about what they mean.

**2.3 Model Spin-Up**

I see divergences of maybe 0.01 K in the plot so either I've completely missed the point or I would suggest they are negligible and insignificant relative to the larger uncertainties you go on to treat.

**3.1 Signal to noise analysis of isotope data**

I wasn't sure what the point of this section was. If it is simply to "evaluate the suitability of the three different inversion targets ($\delta$15N, $\delta$40Ar, $\delta$15Nexcess)" prior to inversion, it would be sufficient to state that these data have been used like this before.

On the other hand, the statements about 15Nexcess as losing information about LID is really interesting but needs more development.

**3.2 Gas-isotope fitting results**

The procedure for the 10 different fittings runs is not properly described. What is varied so that the 10 runs differ (analytical error, accumulation, etc.)? Also, 10 iterations is generally not a large enough suite to arrive at a statistically robust distribution. How sensitive are your

mean and standard deviations to the number of runs?  This is crucial information as it used to then to conclude the ultimate uncertainty in the inversion.

**3.2.2 Final misfits:**

My take home was that the mean solution was within the analytical uncertainty, which is not surprising, given that you'd developed this algorithm to fit the data.  I think these could easily be incorporated into the above sections. A reordering of these lines of argument might help the readability here.

**3.2.3 Boundary effect**

I've struggled a bit to understand this section and why you have a boundary problem starting about 1000 years before the last data point. Perhaps a useful analogy would to consider a present day firn site.  Why would we need information that's 1000 years "in the future" to understand the gas isotopes of today?

Otherwise, **y**our description of the competing effects makes it sound like your system is severely underdetermined and could easily lead to spurious fits elsewhere.

**3.2.4 Influence of different accumulation-rate estimates .**

My first impression from the figure is that the effect of accumulation in a well-known constrained core like GISP2 is small.  Yet on the other hand I missed the specific effect of accumulation on the inversion because the different scenarios are insufficient described.  My questions would be:

-What would erroneously high or erroneously low accumulation impart on the temperature inversion?  Can you robustly define the frequency of centennially-scale variability if an inaccurately variable delta-age is constant stretching or squeezing your periodic signals?  Is the temperature inversion an intractable problem if you don't know accumulation, for example in the deepest sections of the core where the thinning function dominates the annual layer thickness.

This section could be folded into the section about the overall reproducibility (e.g. combined with 3.2.1. and 3.2.2).

**4.1 Solution comparison**

This was the heart of the paper to me. However, I found little utility in the tables with correlation analysis.  The figures are sufficient.

**4.2 Bandpass analysis of isotope data**

This section seems quite tangential or is either misplaced and should go into the raw data analysis.

**4.3 Model comparison**

Just a note on the figures that relates to the discussion of the absolute temperature: having chosen to plot all data as anomalies and thus nearly identical it was confusing to see the subplot with the 2 K temperature difference. I think it would be a safe to state early on that all your temperature estimates are accurate only relative to modern as thus will focus the discussion on anomalies.

**4.5 Comparison of T(δ15N) with the reconstructions of Kobashi et al. 2017 and Buizert et al. 2018**

When rewriting this section please consider the saying "a picture is worth a thousand words". This is the exciting part of the paper but at the moment it reads like description of the figures and tables. Please consider bringing in a broader yet brief discussion of Holocene climate and the implications of your results. Also, a single paragraph over two pages in excessive.

---

## Short Comment (SC1) · 28 Dec 2019

Doring and Leuenberger conducted an important study for testing various reconstruction methods from nitrogen and argon isotope data in trapped air in Greenland ice cores. As the temperature reconstructions from this methodology may provide the most accurate temperature from the past, further advancements are critical. However, the paper is not very well organized and too bulky and very difficult to read, although it may be useful for thesis. Please reorganize and shorten the paper and only provide important figures and findings. Also, it requires further works to get better perspectives as shown below. Therefore, I recommend major revision.

[Figure]

1. For the verification of the method, it is essential to obtain new nitrogen and argon data from other sites in Greenland with similar quality of Kobashi et al. (2008). Many questions asked in this paper can not be answered by simply calculating the same data again and again. Kobashi et al. (2015) provided NGRIP data for the past 2000 years, although the data quality was not as good as Kobashi et al. (2008). Of course, getting new data is not easy, but at lest should discuss what is necessary to clarifies some questions in the future studies.

2. Goujon et al. (2003) and Schwander et al. (1997) are not state of the art models. The study of firn desification should have advanced much further by now. At least, literature review should be provided for what are the current issues on the understanding of firn densification in different time scales, If possible, try to use "the state of art" firn densification model to calculate temperature. In the paper, discussions are very confined in small literatures. Further discussions on the uncertainties of firn densification should be done as well as isotopic fractionations in firn and ice cores.

3. Kobashi et al. (2017)'s method is innovative because it does not directly use firn denfication model to calculate surface temperature (it uses the model for heat diffusion). In addition, only dNexcess is pure temperature proxy. d15N or d40Ar are not temperature proxy if firn model cannot adequately model desification process in the time scale you are concerned. In the paper, it is repeatedly said d15Nexcess is problematic, but it is also possible that firn model is not correct in the time scale of decades to centuries.

4. The paper only uses d15N and d40Ar for temperature, but many temperature proxies are available from the same core or other cores or around Greenland. It is essential to combine all data to draw conclusions. In particular, I strongly recommend to look at the past 1000 years, which has the highest quality in Kobashi et al. (2008) data. Please make a plot Kobashi et al. (2011), d15N temperature reconstruction (your data), borehole temperature reconstruction, and global or European average temperatures for the past 1000 years, etc. If the data is noisy (unreal), it will not correlate with other climate data.

5. Your temperature calculation for d15Nexcess target is very confusing with Kobashi et al. (2017). Kobashi et al. (2017) uses totally different method linking with d15N, So, please clarify that your temperature calculation for d15Nexcess is different from Kobashi et al. (2017).

References: Kobashi, T., Box, J.E., Vinther, B.M., Goto-Azuma, K., Blunier, T., White, J.W.C., Nakaegawa, T., Andresen, C.S., 2015. Modern solar maximum forced late twentieth century Greenland cooling. Geophys. Res. Lett. 42. https://doi.org/10.1002/2015GL064764 Kobashi, T., Kawamura, K., Severinghaus, J.P., Barnola, J.-M., Nakaegawa, T., Vinther, B.M., Johnsen, S.J., Box, J.E., 2011. High variability of Greenland surface temperature over the past 4000 years estimated from trapped air in an ice core. Geophys. Res. Lett. 38. https://doi.org/10.1029/2011GL049444 Kobashi, T., Menviel, L., Jeltsch-Thömmes, A., Vinther, B.M., Box, J.E., Muscheler, R., Nakaegawa, T., Pfister, P.L., Döring, M., Leuenberger, M., Wanner, H., Ohmura, A., 2017. Volcanic influence on centennial to millennial Holocene Greenland temperature change. Sci. Rep. 7. https://doi.org/10.1038/s41598-017-01451-7 Kobashi, T., Severinghaus, J.P., Kawamura, K., 2008. Argon and nitrogen isotopes of trapped air in the GISP2 ice core during the Holocene epoch (0-11,500 B.P.): Methodology and implications for gas loss processes. Geochim. Cosmochim. Acta 72. https://doi.org/10.1016/j.gca.2008.07.006 Goujon, C., Barnola, J.-M. and Ritz, C.: Modeling the densification of polar firn including heat diffusion: Application to close-off characteristics and gas isotopic fractionation for Antarctica and Greenland sites, J. 10 Geophys. Res. Atmos., 108(D24), n/a-n/a, doi:10.1029/2002JD003319, 2003. Schwander, J., Sowers, T., Barnola, J. M., Blunier, T., Fuchs, A. and Malaizé, B.: Age scale of the air in the summit ice: Implication for glacial-interglacial temperature change, J. Geophys. Res., 102(D16), 19483– 19493, doi:10.1029/97JD01309, 1997.

───────────────────────

---

## Editor Comment (EC1) · Eric Wolff (Editor) · 3 Jan 2020

As you know you are now invited to make a final response to all the comments made by the 2 anonymous reviewers and the comment from Dr Kobashi.

Between them the reviewers of the paper are very critical. They make many substantial scientific comments on the paper. In addition they all comment that it is extremely hard to read because of its length and style. While I think you could potentially make changes to respond to most of the scientific issues, I share their view that the paper is not written in a suitable style for a journal article. Please therefore do respond to the comments but do not at this stage invest time in preparing a new version of the paper.

I think it is probable that I will recommend that the paper be rejected to give you a chance to prepare a completely new and more manageable manuscript (However the formal decision comes only after you respond to reviewers).

———————————————

---

## Author Comment (AC1) · 13 Jan 2020

We have attached our replies to the reviews of Anonymous Referee #1, Anonymous-Referee #2 and to the short comment of Dr. Takuro Kobashi as a single supplement pdf document. The reply to the review of Anonymous Referee #2 can be found on pages 1-5, the reply to the review of Anonymous Referee #1 on pages 6-11. The reply to the short comment of Dr. Takuro Kobashi can be found on pages 12-13.

Please also note the supplement to this comment:
https://www.clim-past-discuss.net/cp-2019-132/cp-2019-132-AC1-supplement.pdf

[Figure]

[Figure]

**Supplement:**

We thank reviewer 2 for the detailed examination of the presented work. This allows us to clarify some issues potentially not emphasized enough within our discussion manuscript. Therefore we will use this opportunity to address major issues together with detailed answers to the key points mentioned by the reviewers. Reviewer comments are given in italic letters whereas our replies are given in normal letters.

*Review for "Comparison of Holocene temperature reconstructions based on GISP2 multiplegas-isotope measurements", Döring and Leuenberger, 2019, Climate of the Past Discussions*

*Summary:*

*Döring and Leuenberger present the first application of a new gas-based temperature inversion procedure to previously published data from GISP2 during the Holocene. This is an interesting and timely contribution to the problem of constraining site temperature from histories of inert gas isotopes that has been gaining traction as an alternative to the classic water isotope paleothermoter. The type of technique they describe is particularly useful as it promises a more subjective and quantitative treatment of uncertainty imparted by the inversion.*

*Overall, I was overwhelmed by the volume of information provided in the manuscript. I would challenge the authors to consider a significant reduction in length in a new submission. Considering that a separate, lengthy methods paper has already been published, I was slightly surprised to see the volume of additional information required to establish the robustness of their method.*

Since both reviews as well as Takuro Kobashi short comment suggested shortening the main text, figures and tables we will do this in a revised version. The robustness of the method has already been shown in our method paper (Döring and Leuenberger, 2018). Yet a lot of information concern the uncertainty evaluation of the reconstructed temperature (firn model, fitting algorithm as well as data uncertainties) will remain but moved to a supplementary document, since we are convinced that it is helpful to fully understand what we did.

*The key questions I had after reading the paper are as follows. What information is crucial and to what accuracy must it be known (e.g. accumulation, bore-hole thermometry, water isotopes temperature reconstructions for initialization, etc.)? What are the knowns that go into the inversions to produce estimates of the unknowns? How over or under determined is the system?*

*Additionally, there's a thread that runs throughout the paper about the hazards of using 15Nexcess. Because of the length and wide range of topics, the arguments end up being very dispersed. I would have appreciated a more concise structure to this thread as it took me a number of reads to find all the salient points.*

*Because of the length I have decided not to make line-by-line suggestions. Rather, I've broken the paper down into sections and highlighted what my main "take-home messages" were as a reader and any key questions I had.*

Obviously, we did not manage to guide the reader through all the thoughts we wanted to place in order to clarify the problems arising from using different proxies and models for temperature reconstructions. In our view, such a thorough discussion about signal to noise is yet missing, despite the many publications on this topic.

*Abstract*
*I count five major points in the abstract.*
*1) Comparison of 15N, 40Ar 15Nex*
*2) Gas loss as an artefact in 40Ar*
*3) Comparison of two different densification models*
*4) Treatment of analytical uncertainty*

*5) Comparison of Buizert et al., Kobashi et al, with this study*
*I personally think the paper could be reduced to about five figures along these lines and a few*
*tables; whereas it currently stands at 14 figures and 10 tables.*
*The use of ellipses (…) in the uncertainty on the temperature is unclear to me until you read*
*deep in into the paper (section 4?)*

We thank the reviewer for his outlined structure of the manuscript, which is very helpful for the revision.

*1 Introduction*
*Please define 15N, 40Ar, and crucially 15Nex. Without stating 15Nex is a function of both 15N*
*and 40Ar it is not clear to the reader that you are in some sense doing a joint inversion when*
*you use 15Nex. Additionally, this would be great time to discuss want information you "lose"*
*when using 15Nex (e.g. LID), which is currently buried deep in the section on signal-to-noise.*

*Please remove references to a general body of work as simply "Kobashi et al.," or "his work".*

Yes, we will include the definition in a sentence about the missing information on LID when using d15Nexcess.

*2.1 Firn-models and inversion algorithm*

*This is fine but it tends to re-hash the published methods paper. Also, given that the Goujan*
*and Schwander models actually show small difference when fitting 15N and 40Ar, I'm not*
*sure so much detail is needed about basic things like the different zones of densification and*
*depth resolution. It would suffice to say you simply used the two classic firn models.*

*The point about no geothermal heat flux in the Schwander is an interesting aside –*
*particularly if you were examining a shallow core (<1000 m) - but my take home was that it*
*just imparted a constant offset in the absolute temperature. Given that we are more*
*interested in changes in temperature relative to the modern (when we can actually measure*
*it), I was a bit lost as to why this was so important.*

Based on this information we will shift part of this section into the supplement.

*2.2 Timescale, and necessary data*

*Why do the text and nearly all the figures refer to chronology as the "ice age" of GICC05A?*
*Shouldn't it be the gas-age chronology for the gas-based data?*

*Accumulation-rate input data: the difference in these scenarios needs to be defined. Later on*
*there are many references to the 50km, 100km, etc scenarios but no information about what*
*they mean.*

We used the ice age scale because the measured isotope data is available on this scale at first. The gas age scale has to be modelled using firn models during the fitting procedure. The difference between the two scales is called Delta age. During the fitting procedure with inputs (temperature and accumulation) the Delta ages are adjusted such that the measured isotope data are matched on both scales. Furthermore, we will extent the ,Accumulation-rate input data' section according to Cuffey and Clow (1997).

*2.3 Model Spin-Up*

*I see divergences of maybe 0.01 K in the plot so either I've completely missed the point or I*
*would suggest they are negligible and insignificant relative to the larger uncertainties you go*
*on to treat.*

You may mean 0.01 permil not 0.01 K? The temperature offsets are significantly larger as stated in Figure 2a. Anyhow, we will move this part to the supplement. The spin-up is critical for matching the early Holocene part correctly since there is an inertia in the system that requires the proper spin-up. Furthermore we keep the spin-up constant during the fitting procedure.

**3.1 Signal to noise analysis of isotope data**

*I wasn't sure what the point of this section was. If it is simply to "evaluate the suitability of the three different inversion targets (δ15N, δ40Ar, δ15Nexcess)" prior to inversion, it would be sufficient to state that these data have been used like this before.*

*On the other hand, the statements about 15Nexcess as losing information about LID is really interesting but needs more development.*

We thank to reviewer for this input. Indeed this section is mainly about the suitability of the input targets which has not been discussed in previous publications to the extent we did it here. Yet, we agree that this can be moved to the supplement with a corresponding reference in the main text.
Further, we will extend the discussion about losing the LID information when using 15Nexcess.

**3.2 Gas-isotope fitting results**

*The procedure for the 10 different fittings runs is not properly described. What is varied so that the 10 runs differ (analytical error, accumulation, etc.)? Also, 10 iterations is generally not a large enough suite to arrive at a statistically robust distribution. How sensitive are your mean and standard deviations to the number of runs? This is crucial information as it used to then to conclude the ultimate uncertainty in the inversion.*

The 10 runs corresponds to the fitting algorithm alone, nothing else have been changed. We will add a sentence to this section. Indeed the number of 10 is small but since the deviations between the different runs are small for d15N and d40Ar we did not perform more due to time restrictions. For d15Nexcess the spread is anyway very large, but we did not test whether the range would increase with a higher number of runs.

**3.2.2 Final misfits:**

*My take home was that the mean solution was within the analytical uncertainty, which is not surprising, given that you'd developed this algorithm to fit the data. I think these could easily be incorporated into the above sections. A reordering of these lines of argument might help the readability here.*

Indeed this can be done, yet we actually tried to follow the sub-structure of the method paper, which is not necessary.

**3.2.3 Boundary effect**

*I've struggled a bit to understand this section and why you have a boundary problem starting about 1000 years before the last data point. Perhaps a useful analogy would to consider a present day firn site. Why would we need information that's 1000 years "in the future" to understand the gas isotopes of today?*

*Otherwise, your description of the competing effects makes it sound like your system is severely underdetermined and could easily lead to spurious fits elsewhere.*

The boundary effect restricted to the last roughly 1000 years has to do with the cancelation of thermal and gravitational fractionation as seen in Figure 8e, f. This cancellation effect is mainly driven by small variations in the gas isotope records and slow reaction of the modelled temperature evolution. The fitting procedure leads to a slow drift towards colder, warmer or constant temperature over thousand years resulting in the same modelled isotope values. The correct temperature evolution can only be obtained with further constraints, i.e. borehole temperature or potentially from temporal temperature gradients derived from instrumental data for Greenland (these are maybe to short). Prior to 1000 years this problem is non-existent due to the fact that gas isotope measurements are present that constrain the fitting. Memory effects of the firn column (LID) heights influences modelled gas isotopes on a later point in time. The firn state has an inertia, it cannot quickly be adjusted in the model as well as in reality. If the timing of temperature and firn state changes agrees than

compensation effects develop. This is different for Dansgaard-Oeschger events, where we have rapid temperature changes not allowing this compensation (firn state is too slow to adapt).

**3.2.4 Influence of different accumulation-rate estimates .**

*My first impression from the figure is that the effect of accumulation in a well-known constrained core like GISP2 is small. Yet on the other hand I missed the specific effect of accumulation on the inversion because the different scenarios are insufficient described. My questions would be:*

*-What would erroneously high or erroneously low accumulation impart on the temperature inversion? Can you robustly define the frequency of centennially-scale variability if an inaccurately variable delta-age is constant stretching or squeezing your periodic signals? Is the temperature inversion an intractable problem if you don't know accumulation, for example in the deepest sections of the core where the thinning function dominates the annual layer thickness.*

*This section could be folded into the section about the overall reproducibility (e.g. combined with 3.2.1. and 3.2.2).*

Cuffey and Clow (1997) did look into the range of accumulation rate changes. This resulted in three possible scenarios how the accumulation rate could have changed throughout the Holocene, we named them 50, 100 and 200 km scenarios, following Cuffey and Clow (1997), corresponding to the distance from where the ice at GISP2 today was originating from. We investigated the influence of the accumulation rate changes in the method paper and found it as minor for fast signals. In this study the three different available accumulation rate histories leads to maximal 0.3 K offsets for the fitted gas isotope records. Yet, the accumulation history is not well-defined even not for the Holocene due to the lack of knowledge about the flow path of the ice cored at the GISP2 site.

Whether the accumulation rate was different than one of three scenarios is difficult to say, but what we end up with are Delta ages that are in agreement with measured Delta ages based on layer counting (GICC05 age scale) (see Figure 7).

Indeed it would be very helpful to have additional information about the thinning function, i.e. Delta-depth values for rapid climate shifts like the Dansgaard-Oeschger events, but during the Holocene they are missing. For the glacial period this information is available and therefore it allows adjusting both the temperature and the accumulation rate. This has been shown as proof of concept on NGRIP data in Döring and Leuenberger (2018).

**4.1 Solution comparison**

*This was the heart of the paper to me. However, I found little utility in the tables with correlation analysis. The figures are sufficient.*

Thank you for this judgment about the correlation tables, we delete or move them to a subsection of the supplementary material.

**4.2 Bandpass analysis of isotope data**

*This section seems quite tangential or is either misplaced and should go into the raw data analysis.*

We will partly move this section to the suitability of the target data and the other part will be incorporated into section 4.1.

**4.3 Model comparison**

*Just a note on the figures that relates to the discussion of the absolute temperature: having*

*chosen to plot all data as anomalies and thus nearly identical it was confusing to see the subplot with the 2 K temperature difference. I think it would be a safe to state early on that all your temperature estimates are accurate only relative to modern as thus will focus the discussion on anomalies.*

Thank you, we will include it.

**4.5 Comparison of T(δ15N) with the reconstructions of Kobashi et al. 2017 and Buizert et al. 2018**

*When rewriting this section please consider the saying "a picture is worth a thousand words". This is the exciting part of the paper but at the moment it reads like description of the figures and tables. Please consider bringing in a broader yet brief discussion of Holocene climate and the implications of your results. Also, a single paragraph over two pages in excessive.*

We will shorten and rephrase this paragraph

Reviewer 1:

We thank reviewer 1 for his view of the presented work. This allows us to clarify some issues potentially not emphasized enough within our discussion manuscript. Therefore we will use this opportunity to addresses major issues together with detailed answers to the key points mentioned by the reviewers. Reviewer comments are given in italic letters whereas our replies are given in normal letters.

*Scientific significance: Poor*

*Does the manuscript represent a substantial contribution to scientific progress within the scope of Climate of the Past (substantial new concepts, ideas, methods, or data)?*

*This manuscript uses already published data (d15N and d40Ar at GISP2 over the holocene), but a different algorithm (also already published by the authors) to reconstruct the surface air temperature. Although this algorithm presents some advantages, it does not lead to a better fit to the data, and the temperature reconstruction is not better than the 2 already published, and does not provide any new insight on climate.*

We do agree with the first statement that we are using already published data, but we disagree with the statement that it does not lead to a better fit to the data and that the temperature reconstruction is no better than the two already published ones. We clearly show that our fitting procedure lead to an improved mean deviation from the data and to a smaller uncertainty in the reconstructed temperature as given in the other publications if it exists at all. We also do not agree with the statement that there are no new insights on climate. For instance, there are substantial differences in temperature to already published records, in particular in the mid-Holocene (6 to 5 ky BP), where we observed a clear temperature decrease of about 1 K. In a submitted manuscript to Science Advances we show that this change is in agreement with changes in North Atlantic circulation and marine temperature proxies from the North Atlantic (Döring and Leuenberger, 2019).

*Scientific quality: poor*

*Are the scientific approach and applied methods valid? Are the results discussed in an appropriate and balanced way (consideration of related work, including appropriate references)?*

*The consideration of related work is appropriate. The method is valid in theory, but the implementation has some serious flaws.*

*The problem here is essentially the resolution of a sytem of 2 equations with 2 unknowns. We have 2 datasets : d15N and d40Ar, plus a third one which is the borehole temperature profile, and 2 input parameters: temperature and accumulation rate. The data (d15N and d40Ar) can be computed to derive 2 variables : the lock-in depth (LID) and the firn temperature gradient DeltaT. A firn densification/heat diffusion model can be use to relate the climate variables (T and acc) to the LID and DeltaT. The authors use 2 good densification models, but strangely, they decide not to use both datasets together in the inverse problem. So they are left with an under-determined system. They are proud to fit the data perfectly, but it is always possible with an under-determined system, and they do not find a coherent answer when they fit different targets (d15N, d40Ar, d15Nexcess). This means that the problem is not well posed. If you have data, you should use them. As a result, the reconstructed temperature makes no sense, especially for d15N excess, but also for the others, as is visible in figure 8 g, h, and i, where the output of the inverse model produces solutions that do not match the borehole temperature data, or known LID (which is unfortunately not shown).*

We agree with the reviewer's view when dealing with large glacial variations, but the present study is focusing on the Holocene. Indeed the principle of the two processes is still valid but the changes are significantly smaller than during the glacial period. Therefore, the signal-to-noise ratio is much lower during the Holocene and hence the measurement uncertainty is more of an issue than during large glacial variations. This rather stable

climate state of the Holocene is also mirrored by relatively small changes of the accumulation rate based on annual layer counting. Therefore, the uncertainty in the accumulation rate is rather small during the Holocene due to small variations but also due to precise layer counting. We have used the three different accumulation rate histories as published by Cuffey and Clow (1997) and kept the respective accumulation rate unchanged during the fitting procedure. Hence, we have only the temperature to be constrained and therefore the system is not under-determined as falsely stated by the reviewer. Furthermore, we have demonstrated that the algorithm is also working when using a combined target or both targets (d15N and d40Ar) if necessary, i.e. the large glacial variations (Döring and Leuenberger, 2018 and this study). We agree with the reviewer's statement that if you have data available you should use it, but the question is allowed how to use them. We also use them to show that there is evidence for systematic too high δ40Ar data in the early- and late-Holocene potentially caused by post coring gas-loss or an insufficient correction of this mechanism. If one would use the data per se than such a statement is not possible and may lead to wrong temperature reconstructions. Regarding d15Nexcess, the reconstruction does only make sense once you combine it with either the borehole temperature record or if it is adjusted to match the evolution of the nitrogen isotopes as done by Kobashi et al., (2008). The reviewer mentions known LID values, we would appreciate if the reviewer could send as a record of LID for the Holocene period that is discussed here but to our knowledge no such records exists that is derived from measurements except gas isotopes as used here.

*Presentation quality: fair*

*Are the scientific results and conclusions presented in a clear, concise, and wellstructured way (number and quality of figures/tables, appropriate use of English language)?*

*The text is long, sentences are sometimes very complicated with improper english. I recomment a full rewrite of the paper when the scientific objectives are better defined. General comment on the form of the figures: it would be much better to have the time axis going always the same way. Figure 9 is the worst, with 2 different directions in the subplots of the same figure.*

We agree that the text is long and sentences are sometimes very complicated. We will take corresponding actions to edit the text. Furthermore, we will rearrange the manuscript according to suggestions given by reviewer 2. The axis problem will be removed.

*As a result, I suggest that the paper be rejected, and rewritten for future submission. The work is not entirely uninteresting, but as it stands, it looks like a student project, and does not add to the litterature.*

*If I understand the author's motivation correctly, they wanted to use their inverse method to better quantify the uncertainties in temperature reconstructions from inert gases. This is still a valid goal in my mind, but it needs to be tackled differently from what is being done here.*

*First, the authors should adopt the thinking framework of considering gravitational and thermal fractionation separately, and evaluating both of them. It would avoid the tragedy of Figure 7 or 8h, where inapropriate firn thicknesses are derived.*

We do not understand this statement, since our method does take care of these two fractionation processes separately but following another approach than used by other methods. This is discussed in our method paper (Döring and Leuenberger, 2018). It is not exactly clear to us what the reviewer would like to say about Figure 7 and Figure 8. Inappropriate firn thicknesses are obtained only for the d15Nexcess fitting which indicates that information about LID is lost for this parameter through its definition. Therefore, also Delta ages are out of the measured range as shown in Figure 7.
The spread of LID in Figure 8 is due to a boundary problem as discussed. We do not agree that our calculated LID values are inappropriate, in particular since no measurements are available for LID. Reconstructed LID values are very consistent between d15N and d40Ar reconstructions (see additional graphs below).

*Regarding gravitational fractionation, here are some questions that are worth investigating:*

*1. How well is it known from the data? What is the impact of the uncertainty in gas loss on the LID estimation? What is a reasonnable uncertainty range when a convective zone is considered? The output of such a section would give you a LID with uncertainty bars that represent not only the analytical uncertainty, but also other sources, that are more tricky to quantify.*

The first question is implicitly discussed by equation 4 of the manuscript, since this equation is obtained by solving for the gravitational and thermal diffusion fractionations based on the measured d15N and d40Ar values. The thermal diffusion value is directly obtained by eq. 4, whereas the gravitational part is obtained by subtracting the correspondent thermal fractionations from either d15N or d40Ar. Anyhow, this has been used in many other publications. The correspondent temperature gradient in the firn is compared to our fitting method in Figure 10c and Figure 11c (d15Nexcess fit). An indication of the gas loss influence on LID can be seen from Figure 10b inset. It corresponds to a range of -2 to 1 meter. The convective zone is indeed a difficult issue which can vary in the same order as this gas loss derived LID change, but the consequences would be different. A convective zone would lead to a correspondent change in both isotope ratios (d15N, d40Ar) and hence the fitting would lead to consistent LID estimates in contrast to a gas loss of mainly one gas component (here argon) or any other process that favors one isotope.

The following graph shows the gravitational part for d15N calculated from the measurements (d15N, d40Ar, hybrid) and estimated from both models (Schwander and Goujon). Deviations from the measurements are obtained in the early and late Holocene. In order to match the increase in measured gravitational part of d15N, d40Ar (shown below) a four degrees temperature cooling (Figure 11a) and high LIDs (Figure 7 bottom left) are required. These changes seem unrealistic to us.

[Figure]

*2. Can the densification models accurately reproduce LID variability? This is a big question. Lundin et al 2017 have shown that there are substantial offsets in the mean steadystate LID already, this is why you need a T offset between the 2 models. But what about the variability? If you use a reasonnable range of temperature and accumulation rate change, can you reproduce the high frequency variability in the LID? I bet you will not be able to get a perfect fit, and you'll find that the LID changes more slowly in the model that what the data suggest.*

This is a valid question that we also posed, therefore we compared two models and the agreement was very good. Lundin investigates mainly static not dynamic cases, which is important for reconstructions such as discussed here.
An important point is of course the availability of LID values independent of models and gas isotopes, otherwise we run into a circular argument. The following graphs show that the LID estimates are very similar

for d15N, d40Ar and the hybrid solution but not for the d15Nexcess if the Schwander and Goujon models are compared.

[Figure]

[Figure]

*What is shown on Figure 8 g and H is the compensation between the LID and DeltaT.
So all the uncertainty in our knowledge of densification is automatically compensated
by a temperature adjustment so that thermal fractionation compensates the misfit in
gravitational fractionation when you fit d15N only. This is an essential point, that is
causing you a lot of trouble, but you have not discussed it, or even shown a model data
comparison with LID and DeltaT data. You approach supposes that the firn densification
model is perfect, and that is a difficult assumption to make.*

Frist of all, LID and DeltaT can be calculated from the gas isotope records (d15N and d40Ar), but both are temperature dependent. This makes a temperature reconstruction based on d15Nexcess alone difficult. It requires the information about LID changes as well. This information can be gained by using both isotopes under perfect conditions (no systematic errors in the data) as it is the case in a firn model. Figure 8 g,h show deviations between the runs since the boundary conditions are not known as well as during previous times.

This cancellation effect is mainly driven by small variations in the gas isotope records and slow reaction of the temperature evolution. The fitting procedure leads to a slow drift towards colder, warmer or constant temperature over thousand years resulting in the same modelled isotope values. The correct temperature evolution can only be obtained with further constraints, i.e. borehole temperature or potentially temporal temperature gradients derived from instrumental data for Greenland (these are maybe to short). Prior to 1000 years this problem is non-existent due to the fact that gas isotope measurements are present that guides the fitting. Memory effects of the firn column (LID) heights influence modelled gas isotopes on a later point in time. The firn state has an inertia, it cannot quickly be adjusted in the model as well as in reality. If the timing of temperature and firn state changes agree than compensation effects develop. This is different for Dansgaard-Oeschger events, where we have rapid temperature changes not allowing this compensation (firn state is too slow to adapt). That is why the algorithm cannot distinguish between gravitational and thermal diffusion signals and many solutions are possible due to – as stated by the reviewer – compensation effects between thermal diffusion and gravitational fractionation. The same happens always for d15Nexcess therefore, it drifts off when not corrected for it. This boundary effect is discussed thoroughly in the manuscript.

*Temperature influences both gravitational fractionation (through its influence on firn thickness) and thermal fractionation. Kobashi et al., 2008 have already shown that d15Nexcess alone cannot be faithfully used to reconstruct temperature change, because of the problem with drift. It is not new. Your hybrid solution is a good idea, but why not also use the borehole temperature record, or compare with d18O, which is also sensitive to temperature? Here several options are possible. But what you did does not include enough constraints to give a meaningful result. In theory, your inverse method can be used to fit multiple datasets at the same time. I suggest that you would do that.*

We agree with the reviewer that Kobashi et al. (2008) were aware of the drift problem otherwise they would not have corrected for the drift. Yet, this is a critical issue especially regarding the compensation effect as shown in Figure 7. With a hybrid solution we actually show that Kobashi et al., (2008) results can be matched very well with our approached of a superposition of a long-term signal from d15N and the short-term d15Nexcess signals. It is a proof of concept of our algorithm. However, the results are slightly different mainly due to the fact of different strategies of accounting for the drifts. The corrections of the drifts is a tradeoff between the d15N fit and d15Nexcess fit which can lead to different temperature estimates.

*Finally, there are some interesting differences between the Kobashi and the Buizert temeprature reconstructions, particularly around 8.2ka, where Kobashi has a big overshoot after 8ka, but Buizert does not. It would be interesting to understand where these differences come from. It would also be interesting to see whether using their temperature histories with the two densification models, you find a good fit to the data. By comparing the different reconstructions, and the d18O record, you could have a more in-depth discussion about climate history at GISP2. It would give more meat to the paper, and justify its presence in CP.*

This is discussed in the manuscript.

*Tu sum up, as it stands, this paper does not reach substantial conclusions that justify its publication, but with a bit of work, the authors could submit a new manuscript on this general topic.*

***Takuro Kobashi***

*takuro.kobashi@gmail.com*

*Doring and Leuenberger conducted an important study for testing various reconstruction*
*methods from nitrogen and argon isotope data in trapped air in Greenland ice*
*cores. As the temperature reconstructions from this methodology may provide the*
*most accurate temperature from the past, further advancements are critical. However,*
*the paper is not very well organized and too bulky and very difficult to read, although*
*it may be useful for thesis. Please reorganize and shorten the paper and only provide*
*important figures and findings. Also, it requires further works to get better perspectives*
*as shown below. Therefore, I recommend major revision.*

As suggested also by two reviewers we considerably will change the paper layout and significantly shorten it and rearrange sections. Yet a lot of information will remain but moved to a supplementary document, since we are convinced that it is helpful to fully understand what we did.

*1. For the verification of the method, it is essential to obtain new nitrogen and argon*
*data from other sites in Greenland with similar quality of Kobashi et al. (2008). Many*
*questions asked in this paper can not be answered by simply calculating the same data*
*again and again. Kobashi et al. (2015) provided NGRIP data for the past 2000 years,*
*although the data quality was not as good as Kobashi et al. (2008). Of course, getting*
*new data is not easy, but at lest should discuss what is necessary to clarifies some*
*questions in the future studies.*

New measurements are out of the scope of this study. Of course new data would be helpful in particular on NGRIP where already data exist as you mention. Yet, as documented in our manuscript our method does not allow a robust reconstruction for a short recent period due to the boundary effect if we neglect using the borehole temperature information.

*2. Goujon et al. (2003) and Schwander et al. (1997) are not state of the art models.*
*The study of firn desification should have advanced much further by now. At least, literature*
*review should be provided for what are the current issues on the understanding*
*of firn densification in different time scales, If possible, try to use "the state of art" firn*
*densification model to calculate temperature. In the paper, discussions are very confined*
*in small literatures. Further discussions on the uncertainties of firn densification*
*should be done as well as isotopic fractionations in firn and ice cores.*

Well Goujon and Schwander are often used and still correspond to models of choice in many publications. Of course there are additional models with newer additions of bubble close-off processes, close-off fractionations (Birner et al., 2018). The latter publication document decreased d15N values at close-off in the order of 5 permeg, which is similar to the values as obtained for the short-term variations (multi-decadal) seen in the measured GISP2 data. However, the good agreement between the two models that we use, document that it is most probably not a model issue that limits the temperature and accumulation rate reconstructions.

*3. Kobashi et al. (2017)'s method is innovative because it does not directly use firn denfication*
*model to calculate surface temperature (it uses the model for heat diffusion). In*
*addition, only dNexcess is pure temperature proxy. d15N or d40Ar are not temperature*
*proxy if firn model cannot adequately model desification process in the time scale you*
*are concerned. In the paper, it is repeatedly said d15Nexcess is problematic, but it is*
*also possible that firn model is not correct in the time scale of decades to centuries.*

We agree with this comment to a certain extent, namely that also firn models might be one reason of concern for decadal variations (see above). Yet, tests have shown that the short-term variability of accumulation rates have minor influences on LID and temperature for models.

*4. The paper only uses d15N and d40Ar for temperature, but many temperature proxies*

*are available from the same core or other cores or around Greenland. It is essential
to combine all data to draw conclusions. In particular, I strongly recommend to look
at the past 1000 years, which has the highest quality in Kobashi et al. (2008) data.*

*Please make a plot Kobashi et al. (2011), d15N temperature reconstruction (your data),
borehole temperature reconstruction, and global or European average temperatures
for the past 1000 years, etc. If the data is noisy (unreal), it will not correlate with other
climate data.*

The comparison between the d15N derived temperature profile and the measured borehole temperature profile as shown in Figure 8i is good for those runs that show a cooling trend in recent centuries (blue curves). Based on this comparison we state that the most plausible temperature evolution is a cooling and the steady (green curves) and increasing evolution (red curves) also obtained from the algorithm due to the lack of input information (future) are not compatible with the borehole record. We will try to come up with a comparison for the last 1000 years in the revised version.

*5. Your temperature calculation for d15Nexcess target is very confusing with Kobashi*

*et al. (2017). Kobashi et al. (2017) uses totally different method linking with d15N,
So, please clarify that your temperature calculation for d15Nexcess is different from
Kobashi et al. (2017).*

Temperature reconstructions using d15Nexcess as a target is very dispersive among different fitting runs. This is the case because information about LID gets lost (cancellation of gravitation effect). In the hybrid method we combined the long-term trend  solution (>500 years) based on d15N and the short-term variations in d15Nexcess in order to obtain a solution that is very much similar to the temperature obtained by Kobashi et al., (2017).

Additional References that are not part of the Discussion paper:

Birner, Benjamin, et al. "The influence of layering and barometric pumping on firn air transport in a 2-D model." *The Cryosphere* 12.6 (2018): 2021-2037.